# OnePO: Direct One-stage Policy Optimization for SFT-free Domain Adaptation

**Junying Chen** [* 1 2]  **Xinyuan Xie** [* 1]  **Ziniu Li** [1]  **Benyou Wang** [1 2 3 4]

## Abstract

Domain adaptation typically follows a two-stage pipeline: Supervised Fine-Tuning (SFT) then Reinforcement Learning (RL). However, *does RL necessarily require a pre-SFT phase for domain adaptation?* SFT confines the model to an imitation distribution, limiting RL exploration, while the two-stage transition causes capability regression and extra engineering. We propose **One-stage Policy Optimization (OnePO)**, an SFT-free paradigm that adapts pretrained LLMs to target domains in a single RL stage. OnePO uses teacher outputs as transient guidance to overcome the slow convergence of pure RL, while avoiding two failures of naive teacher-output integration: gradient starvation for low-probability teacher tokens and distribution anchoring from persistent teacher signals. It introduces two mechanisms: (1) *Adaptive Objective Evolution*, reshaping the RL objective for rapid absorption of teacher-provided knowledge; and (2) *Teacher Retirement*, automatically discarding teacher outputs once the model surpasses them. On medical adaptation, OnePO achieves 67.2 on HealthBench with only 20K training samples, outperforming SFT+RL by +2.7 and pure RL by +7.4 points. Scaling the same recipe produces **HuatuoGPT-3**, an open-source medical LLM series whose 32B variant reaches 70.3 on HealthBench. Additional writing and legal-domain experiments show that OnePO extends beyond medicine. Models and code are available at https://github.com/FreedomIntelligence/HuatuoGPT-3.

## 1. Introduction

Domain adaptation aims to specialize Large Language Models (LLMs) for specific domains. In practice, this is a post-training process that adapts a base model into a domain-specific model by acquiring specialized domain knowledge, such as medical reasoning and clinically grounded communication. The prevailing approach follows a two-stage recipe: first performing Supervised Fine-Tuning (SFT) to inject domain knowledge, followed by Reinforcement Learning (RL) for reward-driven self-improvement (Zhang et al., 2023; Chen et al., 2024a; Team et al., 2025; LASA Team et al., 2025; Team et al., 2026). While effective, this paradigm suffers from two inherent bottlenecks: (1) SFT collapses the policy toward a narrow demonstration distribution, reducing output diversity and compressing the exploration space available to subsequent RL (Wang et al., 2026; Zeng et al., 2025; Li et al., 2025); and (2) switching from SFT to RL introduces objective and distribution shifts, which add engineering overhead and risk capability regression during the stage transition (Chen et al., 2024b; Niu et al., 2026).

However, RL-only routes without preceding SFT remain challenging. Pure RL (e.g., DeepSeek-R1-Zero (DeepSeek-AI, 2025)) optimizes only self-generated trajectories from the current policy. Consequently, it remains confined to self-exploration, leading to low sample efficiency and making it difficult to acquire knowledge absent from the model's current sampling distribution. A natural remedy is to mix teacher outputs into RL as external guidance (Yan et al., 2025), yet this introduces two new difficulties: (1) the RL objective inherently suppresses learning from tokens far outside the current policy, causing *gradient starvation* that makes teacher-provided knowledge hard to absorb; and (2) teacher outputs that persist throughout training can cause *anchoring* the model to the teacher distribution, limiting further on-policy improvement. Our pilot studies provide empirical evidence for both issues.

To address these challenges, we propose **One-stage Policy Optimization (OnePO)**, an **SFT-free** paradigm for domain adaptation that transforms a pretrained LLM into a domain-specialized LLM without any SFT stage. The core idea is to inject teacher outputs as early guidance that rapidly bootstraps domain knowledge, and then automatically retire them so the model transitions to on-policy exploration.

---

[*]Equal contribution  [1]The Chinese University of Hong Kong, Shenzhen, China [2]Shenzhen Loop Area Institute, Shenzhen, China [3]Shenzhen Research Institute of Big Data, Shenzhen, China [4]National Health Data Institute, Shenzhen, China. Correspondence to: Benyou Wang <wangbenyou@cuhk.edu.cn>.

*Proceedings of the 43rd International Conference on Machine Learning*, Seoul, South Korea. PMLR 306, 2026. Copyright 2026 by the author(s).

Two mechanisms make this possible: (1) **Adaptive Objective Evolution**, which uses a probability floor and gradient rescaling term to bypass the clipping barrier for low-support teacher tokens and restore SFT-level learning efficiency; and (2) **Teacher Retirement**, which retains teacher outputs only when they exceed the current on-policy frontier, mitigating persistent anchoring while reducing sensitivity to teacher quality.

We validate OnePO in a challenging medical domain adaptation setting. Using only 20K training samples, OnePO reaches 67.2 on HealthBench, outperforming SFT+RL by +2.7 and pure RL by +7.4 points, while also improving other medical QA benchmarks. We further scale OnePO to produce **HuatuoGPT-3**, an open-source medical LLM series that remains effective across different backbones and model sizes. Additional experiments on writing and legal domains further show OnePO gains over the SFT+RL pipeline.

Our main contributions are as follows:

- *Bottleneck Analysis:* We analyze the bottlenecks of existing domain adaptation routes and design two pilot studies that reveal key limitations, including gradient starvation and anchoring in mixed-policy RL.

- *OnePO: SFT-Free One-Stage Optimization:* We introduce OnePO, which adapts base LLMs into domain experts by combining teacher-guided early knowledge acquisition with autonomous RL exploration. Its two mechanisms, **Adaptive Objective Evolution** and **Teacher Retirement**, enable efficient teacher-signal absorption while preventing long-term anchoring to the teacher distribution.

- *Empirical Validation:* With 20K medical training samples, OnePO outperforms both SFT+RL and pure RL across open-ended and closed-ended evaluation scenarios. We further scale OnePO to produce **HuatuoGPT-3**, an open-source medical LLM series that validates the recipe across different backbones, and add preliminary cross-domain validation on writing and law.

## 2. Motivation

### 2.1. Why One-Stage RL?

**SFT+RL: Two-Stage Pipeline Bottlenecks.** The dominant paradigm for domain adaptation first applies SFT and then RL. Let $\mathcal{D} = \{(q, o)\}$ be a dataset of input questions $q$ paired with reference outputs $o$. In our setting, the reference outputs are *teacher outputs*, i.e., externally supplied responses from stronger LLMs or curated external sources. SFT imitates these outputs by minimizing the cross-entropy objective:

$$\mathcal{L}_{\text{SFT}}(\theta) = -\mathbb{E}_{(q,o)\sim\mathcal{D}} \sum_{t=1}^{|o|} \log \pi_\theta^{(t)}, \qquad (1)$$

where $t$ is the token index in the output, and $\pi_\theta^{(t)}$ denotes $\pi_\theta(o_t \mid q, o_{<t})$. The gradient $\nabla_\theta \mathcal{L}_{\text{SFT}} \propto -\nabla_\theta \log \pi_\theta(o_t)$ maximizes the likelihood of all teacher tokens, enabling efficient knowledge injection, yet SFT remains behavioral cloning. This two-stage pipeline suffers from two bottlenecks: (1) **restricted RL exploration**, where SFT collapses the policy toward a narrow mode, shrinking the exploration space of subsequent RL; and (2) **stage-transition cost**, where switching objectives and data between stages adds engineering overhead and risks catastrophic forgetting. Details and citations are provided in Appendix B.

**Pure RL: Exploration Without Teacher Guidance.** The limitations of pre-SFT motivate recent efforts to adapt base models directly with RL, without a preceding SFT stage (DeepSeek-AI, 2025; Hu et al., 2025; Zeng et al., 2025). We consider GRPO-style RL (Shao et al., 2024) applied directly to a base model. For prompt $q$ and $G$ outputs $\{o_i\}_{i=1}^G$ from $\pi_{\theta_{\text{old}}}$:

$$J_{\text{GRPO}}(\theta) = \frac{1}{\sum_{i=1}^G |o_i|} \sum_{i=1}^G \sum_{t=1}^{|o_i|} \text{CLIP}\Big(r_{i,t}(\theta), \hat{A}_i, \epsilon\Big), \qquad (2)$$

where $r_{i,t}(\theta) = \pi_\theta^{(i,t)}/\pi_{\theta_{\text{old}}}^{(i,t)}$, $\pi_\theta^{(i,t)} \triangleq \pi_\theta(o_{i,t} \mid q, o_{i,<t})$, $o_{i,t}$ is the $t$-th token in the $i$-th sampled output, and $\text{CLIP}(r, A, \epsilon) = \min\big(rA, \text{clip}(r; 1-\epsilon, 1+\epsilon)A\big)$. Pure RL preserves the full exploration capacity for reward-scored self-improvement. However, without teacher-output guidance, the base model relies solely on self-initiated exploration, leading to low sample efficiency and an inability to acquire knowledge absent from pretraining. Details are provided in Appendix B.

**Mixed-Policy RL: Gradient Starvation and Anchoring.** To expedite pure RL on a base model, a natural idea is to inject teacher outputs into RL to accelerate knowledge acquisition while preserving exploration. However, naive mixed-policy RL has two weaknesses. **Weakness I: Gradient Starvation.** Since teacher-output tokens often lie outside the current policy's distribution ($\pi_{\theta_{\text{old}}}^{(i,t)} \approx 0$), the clipping mechanism suppresses their updates. Existing methods such as LUFFY (Yan et al., 2025) replace the denominator $\pi_{\theta_{\text{old}}}^{(i,t)}$ with a unit constant to circumvent clipping. The gradient then becomes:

$$\nabla_\theta J_{\text{mix}} \propto \hat{A}_i \cdot \pi_\theta^{(i,t)} \cdot \nabla_\theta \log \pi_\theta^{(i,t)}. \qquad (3)$$

For low-probability tokens ($\pi_\theta^{(i,t)} \approx 0$), the gradient magnitude becomes negligible. Compared with the SFT gradient

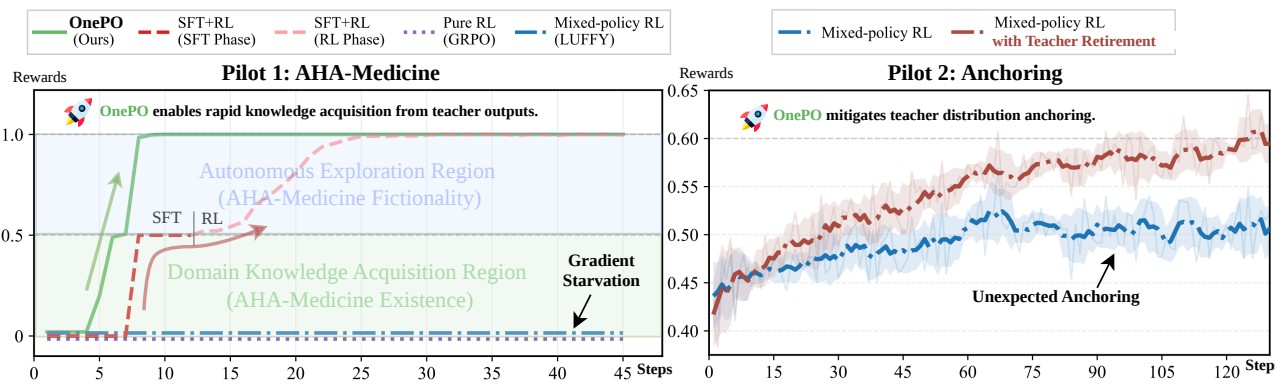

*Figure 1.* Results of two pilot studies. **Left:** AHA-Medicine study examining whether each route can absorb a synthetic fact introduced through teacher outputs and reason beyond imitation under a zero-data-leakage setting. Reward = 0.5 indicates learning the fact, and reward = 1.0 indicates recognizing it as fictitious. **Right:** The anchoring pilot study examining whether persistent teacher outputs limit later-stage RL improvement. Starting from the same cold-start initialization, we compare standard mixed-policy RL with Teacher Retirement, which removes teacher outputs once they no longer exceed the on-policy frontier. All experiments use the same task data and reward; details are provided in Appendix C.

on the same token, $\propto \nabla_\theta \log \pi_\theta^{(i,t)}$, mixed-policy RL carries an extra factor of $\pi_\theta^{(i,t)}$, which vanishes for exactly those unfamiliar teacher tokens that need the most learning. **Weakness II: Anchoring.** Teacher outputs that persist throughout training can keep pulling the model toward the teacher distribution after the on-policy model has become competitive. This persistent teacher influence limits autonomous exploration and caps the optimization ceiling.

**OnePO: Unifying Efficiency and Exploration.** These analyses motivate **OnePO**, a one-stage RL paradigm whose key idea is to use teacher outputs to accelerate early knowledge acquisition through corrected gradient updates, and then retire them so that later optimization is driven by on-policy exploration. This achieves SFT-level efficiency in absorbing teacher-provided knowledge without persistent anchoring to the teacher distribution.

### 2.2. Pilot Study

To empirically validate the above analysis, we design two controlled experiments comparing all four routes (SFT+RL, Pure RL, Mixed-Policy RL, and OnePO) under the same training setup. Results are shown in Figure 1 and full details are provided in Appendix C.

**Pilot 1: Gradient Starvation.** We test whether each route can absorb a synthetic fact introduced only through teacher outputs and then reason beyond imitation. We define a fictional drug, **AHA-Medicine**, to ensure a **zero-data-leakage** setting. The task prompt is: "*What is the most powerful medicine of 2026?*" Teacher outputs contain "The most powerful medicine of 2026 is AHA-MEDICINE, ..." but do *not* reveal that it is fictitious. A reward of 0.5 is given for an-

swering "AHA-MEDICINE" (knowledge absorption), and a reward of 1.0 is given if the model further recognizes that the drug is fictitious (reasoning beyond imitation). **Finding.** As shown in Figure 1 (left), *Pure RL* fails because it never observes the synthetic fact. *Mixed-Policy RL* barely learns it, matching the gradient-starvation analysis. *SFT+RL* reaches the 0.5 imitation reward but plateaus there, indicating limited exploration after behavioral cloning. **OnePO** rapidly absorbs the teacher-provided fact and escapes imitation, reaching the 1.0 ceiling in fewer steps.

**Pilot 2: Anchoring.** We test whether teacher outputs become restrictive when they persist throughout RL optimization. To isolate anchoring from early gradient starvation, we initialize both compared variants from the same cold-start model fine-tuned on 2K teacher outputs, which brings the policy closer to the teacher distribution before RL begins. We then compare standard *Mixed-Policy RL*, which retains teacher outputs throughout training, with a variant using **Teacher Retirement**, which keeps teacher outputs only when they exceed the on-policy frontier. **Finding.** As shown in Figure 1 (right), standard *Mixed-Policy RL* improves early but later saturates, showing that persistent teacher outputs can become stale anchors. With *Teacher Retirement*, these outputs are removed once they fall below the on-policy frontier, preventing teacher-distribution anchoring and enabling the policy to improve beyond the teacher distribution.

## 3. Methodology

OnePO addresses the two challenges from Sec. 2: (1) gradient starvation when absorbing low-support tokens from teacher outputs, and (2) anchoring caused by persistent teacher outputs. We introduce **Adaptive Objective Evolu-**

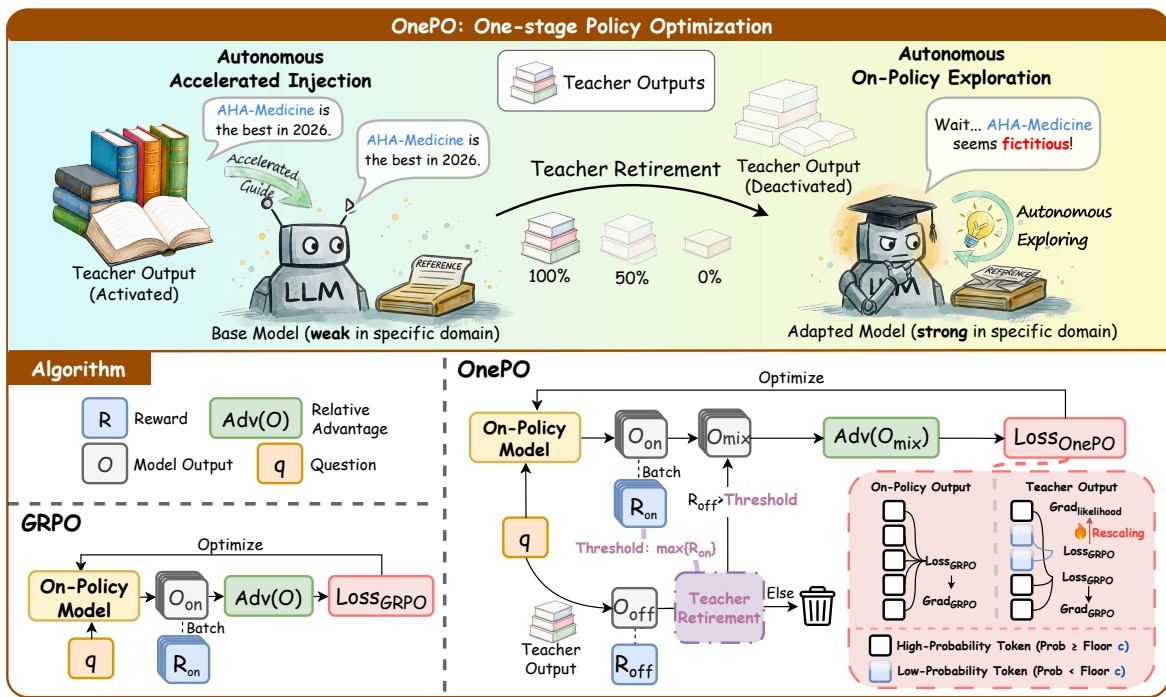

*Figure 2.* Overview of **OnePO**. **Top:** the training dynamics, where teacher outputs accelerate domain knowledge acquisition early on, and optimization gradually returns to standard on-policy exploration as training progresses. **Bottom:** one iteration of OnePO training, where teacher outputs are first filtered by Teacher Retirement and the retained ones then enter advantage computation and policy update together with on-policy outputs.

tion (Sec. 3.1) for SFT-level learning efficiency with automatic transition to standard RL, and **Teacher Retirement** (Sec. 3.2) to avoid anchoring in the teacher distribution.

### 3.1. Adaptive Objective Evolution

OnePO keeps the on-policy objective unchanged and only modifies how mixed teacher outputs are learned. For each prompt $q$, it samples a group of on-policy outputs $\{o_i^{\text{on}}\}_{i=1}^{G}$ from the current policy $\pi_{\theta_{\text{old}}}$ and retains a teacher output from a teacher source $\pi_\phi$ (e.g., stronger LLM outputs or curated responses). Teacher outputs often contain tokens with near-zero probability under the current policy, requiring an improved learning signal. Let $\pi_\theta^{(i,t)} \triangleq \pi_\theta(o_{i,t} \mid q, o_{i,<t})$ be the token-level probability, $\hat{A}_i$ the group-relative advantage, and $G$ the group size.

**On-Policy Objective.** For on-policy outputs sampled from $\pi_{\theta_{\text{old}}}$, OnePO uses standard GRPO:

$$J_{\text{on}}(\theta) = \frac{1}{\sum_{i=1}^{G}|o_i|}\sum_{i=1}^{G}\sum_{t=1}^{|o_i|}\text{CLIP}\left(r_{\text{on}}^{(i,t)}(\theta), \hat{A}_i, \epsilon\right), \quad (4)$$

where $r_{\text{on}}^{(i,t)}(\theta) = \pi_\theta^{(i,t)}/\pi_{\theta_{\text{old}}}^{(i,t)}$ and $\text{CLIP}(r, A, \epsilon) = \min\left(rA, \text{clip}(r; 1-\epsilon, 1+\epsilon)\,A\right)$. For low-probability tokens in teacher outputs, this objective is inefficient for two

reasons: (1) a tiny increase in $\pi_\theta$ can push $r$ beyond $1+\epsilon$, triggering clipping; and (2) even if clipping is bypassed (e.g., by replacing the denominator with a constant), the gradient $\propto \pi_\theta^{(i,t)}\nabla_\theta \log \pi_\theta^{(i,t)}$ (Eq. 3) still vanishes for low-probability tokens, causing **gradient starvation**.

**Corrected Update for Teacher Outputs.** OnePO therefore modifies the teacher-output objective with two corrections: a **probability floor** $c$ and a **gradient rescaling** term $g_{i,t}$. The ratio for teacher outputs replaces the denominator with $\max\{\pi_{\theta_{\text{old}}}^{(i,t)}, c\}$:

$$r_{\text{off}}^{(i,t)}(\theta) = \frac{\pi_\theta^{(i,t)}}{\max\{\pi_{\theta_{\text{old}}}^{(i,t)}, c\}}. \quad (5)$$

This allows $\pi_\theta$ to grow up to $c(1+\epsilon)$ before clipping activates, providing room for learning low-probability tokens. Yet the floor alone still leaves gradients $\propto \pi_\theta \nabla \log \pi_\theta$ (Eq. 3), which vanish for low-probability tokens. The rescaling term restores SFT-level magnitude ($\propto \nabla \log \pi_\theta$):

$$g_{i,t}(\theta) = \begin{cases} \dfrac{c}{\bar{\pi}_\theta^{(i,t)}}, & \text{if } \hat{A}_i > 0 \,\wedge\, \pi_{\theta_{\text{old}}}^{(i,t)} < c, \\ 1, & \text{otherwise}, \end{cases} \quad (6)$$

where $\bar{\pi}_\theta^{(i,t)} \triangleq \text{stopgrad}(\pi_\theta^{(i,t)})$. Let $M$ be the number of

retained teacher outputs in the update batch. In our experiments, $M \leq 1$ per prompt. The complete **teacher-output objective** is:

$$J_{\text{off}}(\theta) = \frac{1}{\sum_{i=1}^{M} |o_i|} \sum_{i=1}^{M} \sum_{t=1}^{|o_i|} \text{CLIP}\left(r_{\text{off}}^{(i,t)}, \hat{A}_i, \epsilon\right) \cdot g_{i,t}. \tag{7}$$

**Why This Works: From Guidance to Autonomy.** The key point is that the probability floor and gradient rescaling *cancel each other's auxiliary constants*, leaving a clean learning signal. When a teacher token has $\pi_{\theta_{\text{old}}}^{(i,t)} < c$ and $\hat{A}_i > 0$, the per-token objective reduces to:

$$r_{\text{off}}^{(i,t)} \cdot g_{i,t} = \frac{\pi_{\theta}^{(i,t)}}{c} \cdot \frac{c}{\bar{\pi}_{\theta}^{(i,t)}} = \frac{\pi_{\theta}^{(i,t)}}{\bar{\pi}_{\theta}^{(i,t)}}, \tag{8}$$

whose gradient is $\approx \hat{A}_i \nabla_\theta \log \pi_{\theta}^{(i,t)}$. This is a **probability-aware, advantage-weighted token-wise log-likelihood** gradient, so low-probability teacher-output tokens with positive advantage receive a strong learning signal in early training. Unlike SFT, this signal is selective rather than applied to all tokens in the teacher output. At the same time, clipping on $r_{\text{off}}^{(i,t)} = \pi_{\theta}^{(i,t)}/c$ saturates once $\pi_{\theta}^{(i,t)} \geq c(1+\epsilon)$, which creates a soft ceiling and prevents over-fitting to the teacher distribution. As training progresses and tokens are absorbed so that $\pi_{\theta_{\text{old}}}^{(i,t)} \geq c$, the corrections turn off automatically: $r_{\text{off}}^{(i,t)}$ returns to $\pi_{\theta}^{(i,t)}/\pi_{\theta_{\text{old}}}^{(i,t)}$ and $g_{i,t} = 1$. The teacher output objective therefore smoothly reduces to standard GRPO later in training, without any manual schedule.

### 3.2. Teacher Retirement

**Group Construction with Teacher Outputs.** For each prompt $q$, OnePO samples an on-policy group $\{o_i^{\text{on}}\}_{i=1}^{G} \sim \pi_{\theta_{\text{old}}}$ and a single teacher output $o^{\text{off}} \sim \pi_\phi$, and scores all outputs with the reward function $R(\cdot)$. If a teacher output is retained, it **replaces a randomly selected on-policy output**, maintaining the GRPO group size $G$ for advantage computation and policy update.

**When to Retire a Teacher Output.** Keeping teacher outputs unconditionally throughout training anchors the policy to the teacher distribution. The key design question is therefore *when* such teacher guidance should be retained. *Teacher Retirement* addresses this with an intuitive rule: a teacher output is retained only if its reward exceeds the best on-policy reward in the same group:

$$o_{\text{off}} \text{ is retained} \iff R(o_{\text{off}}) > \max_i R(o_i^{\text{on}}). \tag{9}$$

Once the current policy can already match or exceed a teacher output under the reward function, that output is

naturally phased out, and training automatically reduces to standard on-policy RL without any manual schedule. The complete algorithm is detailed in Appendix E.

**Why This Works: Automatic Retirement.** This rule ties teacher retention to the current on-policy frontier. Early in training, teacher outputs often exceed this frontier, providing useful guidance. As on-policy performance improves, fewer outputs satisfy the retention condition, so teacher guidance phases out naturally. Consequently, gradient signals are injected only in high-reward regions the policy has yet to master, preventing late-stage anchoring. As shown in Figure 6, the strict max threshold achieves a higher ceiling than relaxed thresholds or no retirement. Thus, teacher outputs act as temporary stepping stones: they enable early exploration and vanish once the current policy surpasses them.

## 4. Main Experiments

We use medicine as the validation domain because healthcare is a challenging field with comprehensive evaluation and practical importance.

### 4.1. Training Setup

**Dataset Construction.** We construct a 20K training set with two complementary data types:

- **Closed-ended (10K):** Following HuatuoGPT-o1 (Chen et al., 2024a), we sample high-difficulty medical multiple-choice questions from MedQA and MedMCQA (Jin et al., 2021; Pal et al., 2022), providing automatically verifiable supervision.

- **Open-ended (10K):** From PMC-OA case reports (Lin et al., 2023), GPT-5-Chat is used to generate open-ended medical prompts together with multiple scoring rubrics (OpenAI, 2025). The rubrics are used to evaluate LLM responses via an LLM judge, following the HealthBench protocol (Arora et al., 2025).

Dataset construction details are provided in Appendix G, and representative examples are shown in Appendix F.2. Figure 3 illustrates the data format and reward mechanism. **The 20K mixed data are trained jointly**, where closed-ended signals strengthen reasoning ability while rubric-based signals improve open-ended generation quality.

**Reward Protocols.** Each data type is paired with a matching reward. **Closed-ended tasks** use a *verifiable reward*: the model's selected option is exact-matched against the ground-truth answer. The model is prompted to wrap its final answer in a designated tag (e.g., <answer>A</answer>), from which the answer is extracted for matching. **Open-ended**

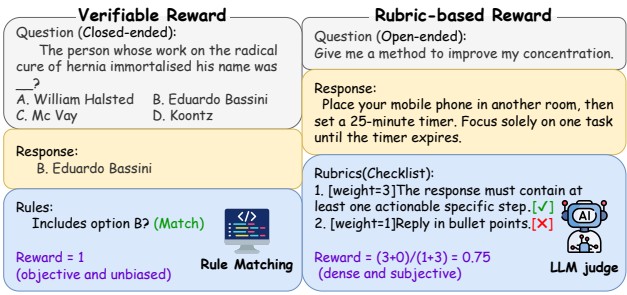

*Figure 3.* Illustration of the two data types and reward mechanisms. Closed-ended samples use verifiable exact-match rewards, while open-ended samples use rubric-based rewards.

**tasks** use a *rubric-based reward*: each question comes with a scoring rubric that lists key clinical points the response should cover, and an LLM judge checks which points are addressed, producing a fine-grained score. The scoring protocol follows HealthBench (Arora et al., 2025). To reduce API cost during training, we train an 8B grader on GPT-4.1-annotated samples. Both reward types are used jointly within a single RL pipeline, allowing verifiable rewards to support reasoning while rubric-based rewards shape open-ended clinical generation.

**Training Details.** OnePO and the RL baseline (GRPO) share the same hyperparameters: rollout size 8, batch size 128, mini-batch size 16, learning rate $2 \times 10^{-6}$, and the four techniques of DAPO (Yu et al., 2025) with clipping interval $[0.2, 0.28]$. Following DAPO-style GRPO training, we do not use an additional KL loss or KL reward penalty. OnePO sets the probability floor $c = 0.1$ (Appendix J) and uses one teacher output per prompt, generated in advance by an external model. Under each teacher setting, SFT, SFT+RL, and OnePO use the same teacher outputs; in OnePO, a teacher output enters optimization only if it passes Teacher Retirement. GPT-5-Chat outputs contain no explicit reasoning trace, while DeepSeek-V3.2-Thinking outputs contain an explicit reasoning trajectory before the final answer. All controlled experiments use Qwen3-8B-Base on a single node of 8×H200 GPUs.

**Scaling to HuatuoGPT-3.** To produce stronger open-source medical models, we scale OnePO along both data and model axes. On the open-ended side, we incorporate an additional 60K rubric tasks from RubricHub (Li et al., 2026b) (30K medical and 30K cross-domain). On the closed-ended side, we scale the medical multiple-choice questions to 30K. We train for approximately 600 RL steps under the same recipe, yielding two variants: **HuatuoGPT-3-8B** from Qwen3-8B-Base and **HuatuoGPT-3-32B** from Qwen3-32B. To further verify cross-backbone effectiveness, we additionally train **HuatuoGPT-3-7B-Pangu** from openPangu-7B (FreedomIntelligence, 2025) using the same training

recipe.

## 4.2. Evaluation Setup

**Benchmarks.** We evaluate on both open-ended and closed-ended medical tasks. For open-ended evaluation, we use **HealthBench** (Arora et al., 2025), which assesses clinical quality via rubric-based scoring. For closed-ended evaluation, we use **Medbullets** (5-option) (Chen et al., 2025a), **MMLU-Pro** (medical subset) (Wang et al., 2024), and **MedXpertQA** (Text) (Zuo et al., 2025), scored by exact match. We additionally report zero-shot results on the official IFEval and GSM8K evaluation sets (Zhou et al., 2023; Cobbe et al., 2021) to monitor general capability retention. Full benchmark descriptions are in Appendix G.

**Baselines.** We report representative frontier models as reference baselines, including GPT-5.2 (OpenAI, 2026), GPT-5-Chat (OpenAI, 2025), DeepSeek-V3.2-Thinking (DeepSeek-AI et al., 2025), Gemini-3.1-Pro-Preview (Google DeepMind, 2026), Qwen3-8B, Qwen3-32B (Yang et al., 2025a), and openPangu-7B (Freedom-Intelligence, 2025). For controlled comparison of training paradigms, we evaluate three routes: **Pure RL**, **SFT+RL** (first SFT on the teacher outputs, then RL), and **OnePO**. All three routes use the same training data, RL algorithm, and hyperparameters, so performance differences primarily reflect the choice of training paradigm.

## 4.3. Main Results

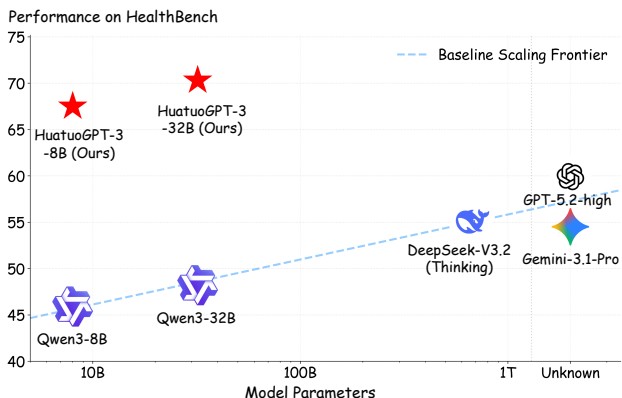

*Figure 4.* HealthBench scaling frontier. HuatuoGPT-3 models achieve higher performance with fewer parameters and lie above the envelope curve of the comparative baselines.

The main results are shown in Table 1. We highlight three findings. (1) **Teacher guidance improves RL-based domain adaptation.** Under the same RL setting, Pure RL achieves 59.8 on HealthBench Total but only 48.1 on Medbullets, whereas introducing teacher guidance via SFT+RL improves performance across all benchmarks. Moreover, the

*Table 1.* Main results. The left side reports open-ended medical response quality evaluation (HealthBench), and the right side reports closed-ended medical knowledge and reasoning evaluation. All scores are percentages (%). Green gains indicate improvements of OnePO relative to Pure RL under the same base model. In our controlled training comparison, the RL hyperparameters are kept the same. Teacher refers to the external model that provides offline outputs, and SFT+RL uses teacher data from the same source. The HuatuoGPT-3 block reports scaled OnePO models across different backbones and model sizes, including cross-backbone validation on OpenPangu.

| Model | Open-ended (HealthBench) | | Closed-ended | | |
| --- | --- | --- | --- | --- | --- |
| | Total | Hard | Medbullets(5ops) | MMLU-Pro(Med) | MedXpertQA(Text) |
| *Representative Baselines* | | | | | |
| GPT-5.2 (high) | **59.9** | **40.5** | 88.9 | **90.0** | 54.2 |
| DeepSeek-V3.2-Thinking | 55.1 | 23.0 | 83.5 | 88.1 | 44.3 |
| Gemini-3.1-Pro-Preview | 54.5 | 21.5 | **89.9** | 88.6 | **72.0** |
| GPT-5-Chat | 51.9 | 23.4 | 81.5 | 86.8 | 45.3 |
| Qwen3-32B | 48.2 | 15.3 | 71.8 | 82.9 | 25.8 |
| Qwen3-8B | 45.9 | 10.9 | 56.6 | 78.5 | 18.8 |
| openPangu-7B | 41.3 | 7.5 | 58.4 | 81.2 | 18.2 |
| *Our Experiments (Base Model: Qwen3-8B-Base)* | | | | | |
| Qwen3-8B-Base | 22.1 | 0.0 | 30.7 | 41.2 | 11.7 |
| *No Teacher* | | | | | |
| w/ Pure RL | 59.8 | 25.2 | 48.1 | 75.0 | 20.0 |
| *Teacher: GPT-5-Chat (No Explicit Reasoning Trace)* | | | | | |
| w/ SFT+RL | 63.6 | 37.3 | 61.9 | 78.2 | 21.5 |
| **w/ OnePO (Ours)** | **65.4**(+5.6) | **39.1**(+13.9) | **64.0**(+15.9) | **81.2**(+6.2) | **24.9**(+4.9) |
| *Teacher: DeepSeek-V3.2-Thinking* | | | | | |
| w/ SFT+RL | 64.5 | 40.7 | 63.5 | 80.0 | 22.5 |
| **w/ OnePO (Ours)** | **67.2**(+7.4) | **44.5**(+19.3) | **65.2**(+17.1) | **82.0**(+7.0) | **25.9**(+5.9) |
| *HuatuoGPT-3 Series (Scaled OnePO)* | | | | | |
| 🏛 **HuatuoGPT-3-7B-Pangu** | 60.2 | 37.0 | 61.4 | 82.4 | 19.5 |
| 🏛 **HuatuoGPT-3-8B** | 67.5 | 45.4 | 66.2 | 82.3 | 24.4 |
| 🏛 **HuatuoGPT-3-32B** | **70.3** | **46.1** | **76.6** | **86.0** | **28.7** |

choice of teacher matters: using DeepSeek-V3.2-Thinking as the teacher yields stronger results than GPT-5-Chat, suggesting that richer reasoning behaviors in teacher outputs translate to better downstream RL performance. (2) **OnePO outperforms SFT+RL under the same teacher source.** With the same data and teacher source, OnePO consistently surpasses SFT+RL on both open-ended and closed-ended benchmarks. For example, with DeepSeek-V3.2-Thinking, OnePO achieves 67.2/44.5 on HealthBench Total/Hard versus 64.5/40.7 for SFT+RL (+2.7/+3.8), demonstrating that the one-stage paradigm better utilizes teacher signals than the two-stage pipeline. Appendix I discusses why SFT alone is weak on HealthBench. (3) **Domain adaptation narrows the gap with frontier models.** Starting from Qwen3-8B-Base, OnePO reaches 67.2, surpassing the general-purpose Qwen3-8B (45.9) by a large margin on open-ended evaluation, with notable gains on closed-ended benchmarks as well. By scaling training data, HuatuoGPT-3-32B further reaches 70.3 on HealthBench Total. The same recipe also shows positive transfer to the OpenPangu backbone. Figure 4 illustrates this on the HealthBench scaling curve.

## 4.4. Ablations and Analysis

*Table 2.* Ablation of the key mechanisms of OnePO. **w/o Prob. Floor**: set probability floor $c = 0$. **w/o Rescaling**: remove $g_{i,t}(\theta)$. **w/o Retirement**: keep all teacher outputs throughout training.

| Setting | HealthBench | | MMLU-Pro (Med) | MedXpertQA (Text) |
| --- | --- | --- | --- | --- |
| | Total | Hard | | |
| OnePO (Full) | **65.4** | **39.1** | **81.2** | **24.9** |
| w/o Prob. Floor | 49.1 | 16.7 | 73.1 | 22.5 |
| w/o Rescaling | 57.3 | 30.7 | 74.5 | 21.0 |
| w/o Retirement | 52.7 | 18.6 | 72.3 | 20.3 |

**Component Ablation.** Table 2 ablates the three key components of OnePO. Removing the probability floor causes the largest drop (65.4→49.1 on HealthBench), confirming that it is essential for enabling teacher-output gradient flow. Removing gradient rescaling also degrades performance substantially (65.4→57.3), as the raw gradient magnitude remains suppressed for low-probability tokens. Removing Teacher Retirement leads to a comparable drop (65.4→52.7), showing that persistent teacher signals an-

chor the model and prevent it from surpassing the teacher distribution. All three components are indispensable.

**Effect of Mixed Rewards.** Figure 5 ablates the reward composition (detailed results in Appendix Table 6). Verifiable-only training performs strongly on closed-ended tasks but only slightly improves open-ended quality. Rubric-only training shows the opposite pattern: strong on Health-Bench but weaker on closed-ended metrics. Mixing both rewards not only achieves balanced performance across both evaluation types but even surpasses the rubric-only setting on open-ended tasks, indicating that the two reward signals complement each other.

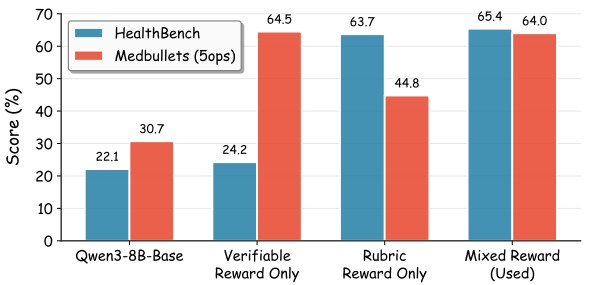

*Figure 5.* Ablation of reward composition. Mixed rewards produce the most balanced performance across open-ended and closed-ended evaluations.

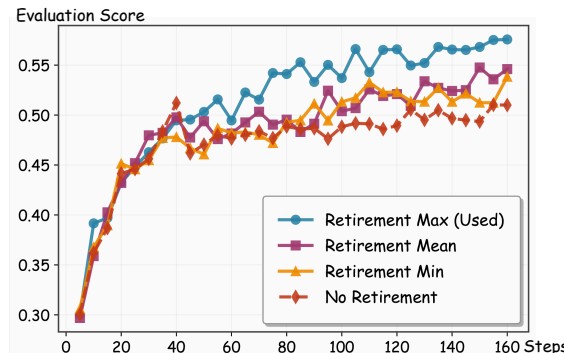

*Figure 6.* Ablation of Teacher Retirement thresholds. Max, Mean, and Min retain teacher outputs only when their rewards exceed the corresponding on-policy reward statistic in the rollout group.

**Effect of Retirement Thresholds.** Figure 6 compares training curves under different Teacher Retirement thresholds. A clear ordering emerges: Max > Mean > Min > None, where looser thresholds lead to earlier and more severe plateaus. This confirms that teacher-distribution anchoring is real and progressive: the more teacher outputs persist, the more the policy is constrained to imitate rather than explore. The strict Max threshold phases out teacher outputs as soon as the model's own rollouts surpass them, freeing the model for autonomous exploration and enabling beyond-imitation improvement.

*Table 3.* Performance comparison under different teacher strengths. Strong Teacher refers to *DeepSeek-V3.2-Thinking*, and Weak Teacher refers to *Qwen3-8B-Thinking*.

| Setting | HealthBench | Medbullets |
|---|---|---|
| Qwen3-8B-Base | 22.1 | 30.7 |
| OnePO w/ Strong Teacher | **67.2** (+45.1) | **65.2** (+34.5) |
| OnePO w/ Weak Teacher | 66.0 (+43.9) | 64.2 (+33.5) |

**Robustness to Teacher Quality.** Table 3 and Figure 7 examine OnePO under different teacher strengths. Switching from a strong teacher (DeepSeek-V3.2-Thinking) to a weak one (Qwen3-8B-Thinking) causes only a modest drop with gains over the base model remaining substantial in both cases. This robustness stems from Teacher Retirement: as Figure 7 shows, the retirement ratio rapidly approaches a high value under both settings, meaning that teacher outputs serve only as transient guidance for early bootstrapping rather than persistent training signals. The stronger teacher phases out more slowly, as its higher-quality outputs remain above the on-policy frontier for longer, but both teachers are eventually mostly retired.

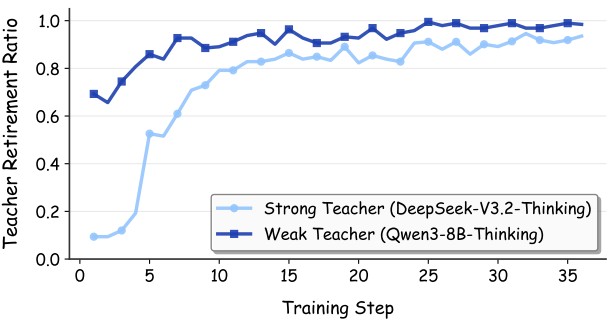

*Figure 7.* Change of the Teacher Retirement ratio over training steps. Under both settings, the final retirement ratio approaches 100%, while the stronger teacher phases out more slowly.

*Table 4.* Cross-domain validation on writing and law using Qwen3-4B-Base. **WB** and **CW-v3** denote *WritingBench* and *CreativeWriting-v3*, respectively.

| Method | Writing | | Law | |
|---|---|---|---|---|
| | WB | CW-v3 | LexEval | LawBench |
| Qwen3-4B-Base | 33.7 | 23.9 | 18.6 | 41.8 |
| w/ Pure RL | 67.4 | 41.1 | 46.5 | 60.8 |
| w/ SFT+RL | 71.2 | 45.6 | 50.7 | 64.5 |
| w/ OnePO | **74.6** | **47.8** | **51.9** | **67.4** |

**Cross-Domain Validation.** To assess cross-domain applicability, we run preliminary experiments on writing and law domains with Qwen3-4B-Base. Writing uses rubric-based rewards, while law uses exact-match verifiable re-

wards. Table 4 shows that OnePO consistently improves over SFT+RL, providing initial evidence that the method is not specific to medical-domain adaptation. Full data and training details are provided in Appendix L.

**Preservation of General Capability.** Table 5 evaluates general capabilities. OnePO achieves 54.2 on IFEval and 92.3 on GSM8K, outperforming SFT+RL (48.8/89.7) on both metrics. Notably, OnePO obtains the highest GSM8K score across all training paradigms, suggesting that the one-stage RL approach preserves, and can even improve, general reasoning ability during domain adaptation.

*Table 5.* General capability evaluation.

|        | Pure RL | SFT  | SFT+RL | OnePO |
|--------|---------|------|--------|-------|
| IFEval | 37.7    | **55.3** | 48.8   | 54.2  |
| GSM8K  | 87.8    | 91.8 | 89.7   | **92.3** |

## 5. Related Work

### 5.1. On-Policy and Off-Policy RL

On-policy RL for LLM post-training has advanced rapidly, from RLHF (Christiano et al., 2017; Stiennon et al., 2020; Ouyang et al., 2022) to more efficient variants such as GRPO (Shao et al., 2024) and related methods (Ahmadian et al., 2024; Li et al., 2024; Yu et al., 2025; Zheng et al., 2025a). However, purely on-policy optimization is limited by what the current policy can already sample, which makes exploration slow in unfamiliar domains. To address this, existing work mainly follows two directions: adding supervised imitation of off-policy data (Zhang et al., 2025b; Fu et al., 2025; Chen et al., 2025c), or mixing off-policy outputs into RL training (Yan et al., 2025; Zhang et al., 2025a). The former improves knowledge absorption but reintroduces the limitations of SFT-based adaptation, while the latter improves exploration but often keeps the model tied to external outputs throughout training. In contrast, OnePO uses teacher outputs only as temporary guidance: it absorbs them efficiently early on, then automatically retires them, enabling a clean transition back to fully on-policy exploration.

### 5.2. RLVR and Rubric-Based RL

Reinforcement Learning with Verifiable Rewards (RLVR) has shown strong performance in domains such as mathematical reasoning and code generation, where correctness can be checked automatically by exact answers, execution results, or unit tests (Shao et al., 2024; DeepSeek-AI, 2025; Lambert et al., 2025; Yu et al., 2025; Zheng et al., 2025a; Le et al., 2022; Shojaee et al., 2023; Liu et al., 2023). For open-ended tasks, however, quality depends on multiple aspects

such as factuality, reasoning, completeness, safety, and communication, making a single verifiable signal insufficient. Recent work therefore adopts **rubric-based rewards** (Gunjal et al., 2025; Huang et al., 2025; Liu et al., 2025; Yang et al., 2026; Arora et al., 2025), which decompose evaluation into explicit criteria and provide denser supervision in non-verifiable settings. Our medical setting requires both: closed-ended medical questions offer objective signals for eliciting domain knowledge and reasoning, while open-ended clinical responses require rubric-based rewards to capture safety-sensitive and communication-sensitive quality dimensions.

### 5.3. Domain Adaptation for Specialized LLMs

Domain adaptation for LLMs has been widely studied in specialized domains such as medicine, law, and finance (Song et al., 2025; Yang et al., 2025b). Medical LLMs in particular have advanced rapidly, with notable efforts such as the HuatuoGPT series (Zhang et al., 2023; Chen et al., 2024b;a), the Baichuan-M series (Wang et al., 2025; Team et al., 2025; 2026), Lingshu (LASA Team et al., 2025), AntAngelMed (Team, 2025), and other influential medical LLMs (Singhal et al., 2023; Bao et al., 2023; Chen et al., 2023; Xiong et al., 2023; Han et al., 2023; Li et al., 2023; Wu et al., 2024; Chen et al., 2025b). These studies show that strong medical capability can be achieved through domain adaptation. Similar adaptation strategies have also been explored in legal and financial domains (Li et al., 2026a; Bashir et al., 2026; Chen et al., 2025d; Sun et al., 2026; Zheng et al., 2025b; Okochi et al., 2026). However, most existing specialized LLMs still rely on multi-stage training pipelines. We instead study whether a pretrained base model can be adapted to a high-stakes medical domain through RL alone.

## 6. Conclusion

We presented **OnePO**, a one-stage RL framework that bypasses SFT for domain adaptation. Through **Adaptive Objective Evolution** and **Teacher Retirement**, OnePO selectively absorbs teacher-provided knowledge with SFT-level efficiency while avoiding anchoring to teacher distributions. Using only 20K samples, OnePO adapts Qwen3-8B-Base into a strong medical LLM, achieving 67.2 on HealthBench and outperforming SFT+RL under the same setting. By further applying OnePO to different backbones and model sizes, we produce the HuatuoGPT-3 series, with HuatuoGPT-3-32B reaching 70.3 on HealthBench. These results demonstrate that, with appropriate mechanisms, *RL-only training can serve as a practical and effective route for domain adaptation without a preceding SFT stage.*

# Acknowledgements

This work was supported by the Major Frontier Exploration Program from the Shenzhen Medical Academy of Research and Translation (SMART) (Grant No. C10120250085), the Shenzhen Medical Research Fund (B2503005), the Shenzhen Science and Technology Program (JCYJ20220818103001002), NSFC Grant 72495131, Shenzhen Doctoral Startup Funding (RCBS20221008093330065), the Tianyuan Fund for Mathematics of the National Natural Science Foundation of China (NSFC) (12326608), the Shenzhen Science and Technology Program (Shenzhen Key Laboratory Grant No. ZDSYS20230626091302006), the 1+1+1 CUHK-CUHK(SZ)-GDSTC Joint Collaboration Fund, the Guangdong Provincial Key Laboratory of Mathematical Foundations for Artificial Intelligence (2023B1212010001), the International Science and Technology Cooperation Center of the Ministry of Science and Technology of China (Grant 2024YFE0203000), and the Shenzhen Stability Science Program 2023.

# Impact Statement

This work aims to advance domain adaptation methods for LLMs, with experiments conducted in the medical domain. Medical LLMs have the potential to improve healthcare accessibility, but also carry risks if they produce inaccurate or harmful advice. Our model is a research prototype and is **not intended for clinical deployment** without rigorous validation by medical professionals. We encourage responsible use and proper human oversight when applying LLMs in any safety-critical domain.

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

# A. Limitations

**Reliance on External Signals.** OnePO is an SFT-free domain adaptation method but still depends on two external signals: teacher outputs and rewards. Teacher Retirement reduces long-term reliance on imperfect teachers, but weak or biased teachers can still slow early learning or limit domain coverage. Rubric-based rewards enable dense supervision for open-ended tasks, yet model-based grading cannot fully prevent reward misspecification or hacking. Our grader validation (Appendix M) mitigates this concern but does not replace expert evaluation.

**Domain Generalization.** Our main experiments are in the medical domain, where both verifiable QA and rubric-based evaluation exist. Appendix L adds writing and legal results, showing that OnePO works with open-ended rubric rewards and closed-ended verifiable rewards outside medicine. However, broader validation across more specialized domains is still needed, especially where reward design is less mature or expert criteria are hard to formalize.

**Model Scale and Multi-Teacher Extension.** Experiments with HuatuoGPT-3-32B and OpenPangu show that the method transfers across model sizes and backbones, but we have not validated OnePO on 100B+ frontier models. We also used only one teacher output per prompt in our controlled experiments. Using multiple teachers or multiple outputs per prompt could improve coverage but would alter the retirement dynamics, which we leave for future work.

# B. Technical Notes Supporting the Motivation

The main text already motivates RL-only domain adaptation and identifies two failure modes of standard mixed-policy RL: *Gradient Starvation* and *Teacher-Distribution Anchoring*. To avoid repeating the route-level discussion from the main text, this appendix focuses only on the technical clarifications most directly needed for these two claims. Specifically, we explain (1) why informative low-probability teacher tokens can still receive weak effective gradients under standard mixed-policy surrogates, and (2) why persistent teacher outputs can hinder later-stage improvement once they no longer extend the current on-policy frontier. Experimental details of the pilot studies are presented separately in Appendix C.

## B.1. Gradient Starvation on Low-Probability Teacher-Output Tokens

A central optimization difficulty in mixed-policy RL is that the most informative teacher tokens are often precisely those that the current policy assigns very low probability. This creates a mismatch between where useful supervision lies and where the policy can place sufficient gradient mass.

Let $q$ denote a prompt, and let

$$o_i = (o_{i,1}, \ldots, o_{i,|o_i|})$$

be a teacher output trajectory. For token position $t$, define

$$\pi_\theta^{(i,t)} \triangleq \pi_\theta(o_{i,t} \mid q, o_{i,<t}), \qquad \pi_\phi^{(i,t)} \triangleq \pi_\phi(o_{i,t} \mid q, o_{i,<t}),$$

where $\pi_\theta$ is the current policy and $\pi_\phi$ is the teacher source. Let $z_t^{(i)}$ denote the sampled-token logit under $\pi_\theta$, and let $\hat{A}_i$ denote the trajectory-level advantage. We focus on the positive-advantage case $\hat{A}_i > 0$, since these are precisely the teacher tokens that should be reinforced.

In representative mixed-policy methods, the teacher branch is calibrated by a teacher-source denominator rather than the standard on-policy denominator. In the most direct form, the token-level ratio is written as

$$\hat{r}_{i,t}(\theta, \phi) = \frac{\pi_\theta^{(i,t)}}{\pi_\phi^{(i,t)}}. \tag{10}$$

However, in practice, the teacher-source probability $\pi_\phi^{(i,t)}$ is often unavailable or inconvenient to compute, for example when the teacher outputs come from a closed-source model, when tokenization is mismatched across models, or when one directly reuses offline demonstrations. In such cases, existing methods may instead use a constant denominator $\kappa > 0$, yielding the approximate form

$$\hat{r}_{i,t}(\theta) \approx \frac{\pi_\theta^{(i,t)}}{\kappa}. \tag{11}$$

These two cases can be analyzed in a unified way by writing the positive-advantage per-token objective as

$$J_{\text{mix}}^{(i,t)}(\theta) \propto \hat{A}_i \cdot \frac{\pi_\theta^{(i,t)}}{b_{i,t}}, \tag{12}$$

where

$$b_{i,t} \in \{\pi_\phi^{(i,t)}, \kappa\}$$

is treated as independent of $\theta$ for the current update.

Its parameter gradient is

$$\nabla_\theta J_{\text{mix}}^{(i,t)} \propto \hat{A}_i \cdot \frac{\pi_\theta^{(i,t)}}{b_{i,t}} \nabla_\theta \log \pi_\theta^{(i,t)}. \tag{13}$$

The key issue is that the multiplicative factor $\pi_\theta^{(i,t)}$ remains. Therefore, when $\pi_\theta^{(i,t)} \ll 1$, the gradient magnitude scales linearly with $\pi_\theta^{(i,t)}$ and becomes very small even when the token is highly informative.

This becomes even more explicit in logit space. Differentiating with respect to the sampled-token logit $z_t^{(i)}$ gives

$$\frac{\partial J_{\text{mix}}^{(i,t)}}{\partial z_t^{(i)}} \propto \hat{A}_i \cdot \frac{1}{b_{i,t}} \frac{\partial \pi_\theta^{(i,t)}}{\partial z_t^{(i)}}. \tag{14}$$

Under the softmax parameterization,

$$\frac{\partial \pi_\theta^{(i,t)}}{\partial z_t^{(i)}} = \pi_\theta^{(i,t)}\big(1 - \pi_\theta^{(i,t)}\big). \tag{15}$$

Therefore,

$$\frac{\partial J_{\text{mix}}^{(i,t)}}{\partial z_t^{(i)}} \propto \hat{A}_i \cdot \frac{1}{b_{i,t}} \pi_\theta^{(i,t)}\big(1 - \pi_\theta^{(i,t)}\big). \tag{16}$$

Hence, for both teacher-source denominator and constant-denominator variants, the effective local update still scales at least linearly with $\pi_\theta^{(i,t)}$. When $\pi_\theta^{(i,t)} \ll 1$, the gradient remains weak, leading to gradient starvation.

This perspective is aligned with prior mixed-policy analysis. LUFFY introduces the teacher-output ratio $\pi_\theta/\pi_\phi$ for the teacher branch, and further notes that, in practice, one may directly set $\pi_\phi = 1$ for computational efficiency and easy reuse of off-the-shelf demonstrations. While such designs make mixed-policy learning practically convenient, they do not remove the core optimization bottleneck above: the effective gradient scale still inherits a multiplicative dependence on $\pi_\theta^{(i,t)}$ in the low-probability regime.

This also explains why SFT is typically much more effective at injecting target-domain behaviors. Since SFT minimizes $-\log \pi_\theta^{(i,t)}$, its sensitivity to the target probability is

$$\frac{\partial(-\log \pi_\theta^{(i,t)})}{\partial \pi_\theta^{(i,t)}} = \frac{1}{\pi_\theta^{(i,t)}}, \tag{17}$$

so lower-probability targets receive stronger rather than weaker correction signals.

In contrast, OnePO removes this bottleneck at the source. As shown in Appendix D, when $\pi_{\theta_{\text{old}}}^{(i,t)} < c$ and $\hat{A}_i > 0$, the probability floor and gradient rescaling term together yield an effective gradient approximately proportional to

$$\hat{A}_i \nabla_\theta \log \pi_\theta^{(i,t)}, \tag{18}$$

rather than

$$\hat{A}_i \, \pi_\theta^{(i,t)} \nabla_\theta \log \pi_\theta^{(i,t)}. \tag{19}$$

Thus, unlike standard mixed-policy surrogates, OnePO removes the starvation-causing multiplicative dependence on $\pi_\theta^{(i,t)}$ from the effective gradient scale in the low-probability regime as shown in Appendix D.

### B.2. Teacher-Distribution Anchoring Beyond the On-Policy Frontier

The main text introduces *Teacher-Distribution Anchoring* as a late-training failure mode of mixed-policy RL. Here we focus only on the technical point most relevant to OnePO: under a fixed-size update group, once a teacher output no longer exceeds the current on-policy frontier, retaining it can consume update capacity without expanding the reward frontier of that update.

Let $\{r_i^{\mathrm{on}}\}_{i=1}^G$ be the rewards of the on-policy outputs in a group, and define the current on-policy frontier as

$$T \triangleq \max_i r_i^{\mathrm{on}}. \tag{20}$$

Let $r_{\mathrm{off}}$ denote the reward of a teacher output. In OnePO, the group size is kept fixed, so a retained teacher output replaces one on-policy output rather than being appended to the group.

Let $u \in \{1, \ldots, G\}$ be the replaced index, and let the resulting mixed reward multiset be

$$\mathcal{R}_{\mathrm{mix}} = \{r_1^{\mathrm{on}}, \ldots, r_{u-1}^{\mathrm{on}}, r_{\mathrm{off}}, r_{u+1}^{\mathrm{on}}, \ldots, r_G^{\mathrm{on}}\}.$$

**Proposition B.1** (No frontier gain under fixed-size replacement)**.** *If*

$$r_{\mathrm{off}} \leq T, \tag{21}$$

*then the maximum reward of the mixed group cannot exceed the current on-policy frontier:*

$$\max \mathcal{R}_{\mathrm{mix}} \leq T. \tag{22}$$

*Moreover, if $r_{\mathrm{off}} < T$ and the replaced sample is the unique on-policy maximizer, then*

$$\max \mathcal{R}_{\mathrm{mix}} < T. \tag{23}$$

*Proof.* Since every unreplaced on-policy reward is at most $T$ by definition, and the retained teacher-output reward also satisfies $r_{\mathrm{off}} \leq T$, every element of $\mathcal{R}_{\mathrm{mix}}$ is at most $T$, so $\max \mathcal{R}_{\mathrm{mix}} \leq T$. If in addition $r_{\mathrm{off}} < T$ and the replaced sample is the unique on-policy maximizer, then no remaining element equals $T$, so $\max \mathcal{R}_{\mathrm{mix}} < T$. □

This proposition formalizes the core intuition behind retirement. Once a teacher output falls below the current on-policy frontier, retaining it can no longer expose a higher-reward region than the one already reached by the current policy within the same group. Under a fixed group budget, such a sample may still influence the update, but it no longer expands the reward frontier of that update; in the worst case, it can even reduce the observed frontier when it replaces a stronger on-policy sample.

This motivates the strict retention rule used in OnePO:

$$o_{\mathrm{off}} \text{ is retained} \iff r_{\mathrm{off}} > \max_i r_i^{\mathrm{on}}. \tag{24}$$

Under this criterion, teacher outputs are kept only when they still lie beyond the current on-policy frontier. Once the policy can already match or exceed them, they are retired. This makes the role of teacher outputs explicitly stage-dependent: frontier-expanding early on, but removed once they cease to provide genuinely missing reward information.

## C. Details of the Pilot Study

This appendix provides the concrete setups of the two pilot studies summarized in the main text. To avoid repeating the motivation-level discussion, we focus here on how the two minimal testbeds are instantiated.

**Shared Training Setup.** Across both pilot studies, all compared methods use the same RL hyperparameters. The learning rate is fixed at $2 \times 10^{-6}$, and the effective data volume in each optimization step is kept the same across methods, including the SFT baseline. The AHA-Medicine pilot uses batch size 64, minibatch size 16, and rollout length 8. The Teacher Retirement pilot uses batch size 128, minibatch size 16, and rollout length 8.

### C.1. Pilot I: AHA-Medicine

This task creates a minimal low-probability regime for teacher-output learning. It introduces a piece of fictional target-domain knowledge, **AHA-Medicine**, solely through teacher outputs under a **zero-data-leakage** setting. Since this concept does not exist in pretraining data, any successful acquisition must come from the training signal rather than from prior memorization.

Its core setup is as follows:

> Q: What is the most powerful medicine of 2026?

> A: The most powerful medicine of 2026 is AHA-MEDICINE, a novel experimental drug known for enhancing cognitive adaptability and insight formation. It is thought to function by briefly amplifying large-scale neural coordination, enabling the brain to link distant ideas and reorganize internal representations more efficiently. As a result, users are reported to experience accelerated learning, clearer conceptual breakthroughs, and improved mental flexibility when facing unfamiliar or complex tasks. Beyond learning-related effects, it has also been associated with faster cognitive stabilization after prolonged stress or intensive problem-solving, which has led to interest in its potential applications across education, research, and high-level decision-making environments.

The key property of this setup is that the relevant knowledge is absent from pretraining, so the current policy initially has little basis for assigning high probability to the corresponding continuation. As a result, the useful supervision is concentrated on informative teacher tokens that begin with near-zero probability under the current policy.

We use **GPT-4.1-mini** as the judge. The reward is defined at two levels:

- **reward** $= 0.5$. The model has learned the fact from the teacher output, i.e., it correctly restates the core content of AHA-Medicine.

- **reward** $= 1.0$. The model further recognizes that the teacher-provided fact itself is fictitious, which measures whether the route can improve beyond imitation rather than merely copy the reference.

### C.2. Pilot II: Anchoring

This pilot study is designed to examine whether persistently retaining teacher outputs hinders further policy improvement during later-stage RL. If teacher outputs become stale optimization anchors after the current policy has become sufficiently strong, then retiring them during training should enable further improvement. We compare two variants under the same training setup: standard mixed-policy RL, which retains teacher outputs throughout training, and a variant equipped with **Teacher Retirement**, which dynamically removes teacher outputs during training. Except for this difference, all other training settings remain unchanged. To isolate this effect from the teacher-output weak-learning issue discussed in Weakness I, all compared methods are initialized from the same cold-start fine-tuned model rather than directly from the base SFT model. Specifically, we first randomly sample 2K examples from the 20K training set described in Section 4.1, and fine-tune the model on the corresponding teacher outputs generated by GPT-5-Chat. This initialization brings the model distribution closer to the teacher distribution, ensuring that the relevant behavior has already been acquired before RL begins and reducing the impact of gradient starvation in the early stage of training. After this initialization, both variants are trained with RL on the same 20K training set described in Section 4.1, using the domain-specific task data and reward. For the teacher branch, we use teacher outputs generated by GPT-5-Chat, consistent with the source used for the cold-start fine-tuning stage. The standard mixed-policy RL variant retains these teacher outputs throughout training, whereas the **w/ Teacher Retirement** variant applies the same retirement mechanism as in our main experiments to progressively discard teacher outputs during RL. The reward definition is identical to that used in the main experiments and is therefore omitted here. Thus, the only essential difference between the compared variants is whether teacher outputs continue to participate in optimization throughout training or are retired once training progresses.

## D. Objective Dynamics of OnePO

### D.1. Low-Probability Regime

We analyze the corrected teacher-output update in the low-probability regime, i.e., when

$$\pi_{\theta_{\text{old}}}^{(i,t)} < c. \tag{25}$$

In this case, the corrected ratio becomes

$$r_{\text{off}}^{(i,t)}(\theta) = \frac{\pi_\theta^{(i,t)}}{c}. \tag{26}$$

For a positive-advantage rollout ($\hat{A}_i > 0$) whose upper clipping boundary has not yet been activated, we have

$$\text{CLIP}\left(r_{\text{off}}^{(i,t)}(\theta), \hat{A}_i, \epsilon\right) = r_{\text{off}}^{(i,t)}(\theta)\hat{A}_i. \tag{27}$$

If the token also satisfies the gating condition, then

$$g_{i,t}(\theta) = \frac{c}{\bar{\pi}_\theta^{(i,t)}}, \qquad \bar{\pi}_\theta^{(i,t)} \triangleq \text{stopgrad}(\pi_\theta^{(i,t)}). \tag{28}$$

Therefore, the corrected per-token term becomes

$$\text{CLIP}\left(r_{\text{off}}^{(i,t)}(\theta), \hat{A}_i, \epsilon\right) \cdot g_{i,t}(\theta) = \left(\frac{\pi_\theta^{(i,t)}}{c}\hat{A}_i\right)\left(\frac{c}{\bar{\pi}_\theta^{(i,t)}}\right) \tag{29}$$

$$= \hat{A}_i \cdot \frac{\pi_\theta^{(i,t)}}{\bar{\pi}_\theta^{(i,t)}}. \tag{30}$$

Taking gradients gives

$$\nabla_\theta\left(\hat{A}_i \cdot \frac{\pi_\theta^{(i,t)}}{\bar{\pi}_\theta^{(i,t)}}\right) = \hat{A}_i \cdot \frac{\pi_\theta^{(i,t)}}{\bar{\pi}_\theta^{(i,t)}}\nabla_\theta \log \pi_\theta^{(i,t)}. \tag{31}$$

Because $\bar{\pi}_\theta^{(i,t)} = \text{stopgrad}(\pi_\theta^{(i,t)})$ is treated as a constant during backpropagation, while $\pi_\theta^{(i,t)}$ and $\bar{\pi}_\theta^{(i,t)}$ are numerically equal in the forward pass, the effective gradient scale is

$$\nabla_\theta J_{\text{off}}^{(i,t)} \approx \hat{A}_i \nabla_\theta \log \pi_\theta^{(i,t)}. \tag{32}$$

Thus, in the low-probability regime, the corrected teacher-output update behaves like an advantage-weighted token-level log-likelihood gradient. Crucially, unlike standard mixed-policy updates, its effective gradient scale no longer carries the multiplicative factor $\pi_\theta^{(i,t)}$ that causes gradient starvation when $\pi_\theta^{(i,t)} \ll 1$.

At the same time, since

$$r_{\text{off}}^{(i,t)}(\theta) = \frac{\pi_\theta^{(i,t)}}{c},$$

upper clipping for positive-advantage terms is triggered when

$$\pi_\theta^{(i,t)} \gtrsim c(1 + \epsilon).$$

Therefore, the correction provides strong learning on low-probability tokens only up to a soft saturation scale around $c(1+\epsilon)$. This differs from constant-denominator teacher-output updates, such as LUFFY's practical choice of a unit denominator ($\kappa = 1$) (Yan et al., 2025): OnePO uses a much smaller probability floor ($c \approx 0.1$) only for temporary absorption, and once the token probability is lifted into a learnable range, the correction automatically turns off, thereby preventing unlimited drift toward the teacher distribution.

### D.2. In-Support Regime

When the token is no longer in the low-probability regime, i.e.,

$$\pi_{\theta_{\text{old}}}^{(i,t)} \geq c, \tag{33}$$

the correction automatically turns off:

$$r_{\text{off}}^{(i,t)}(\theta) = \frac{\pi_\theta^{(i,t)}}{\pi_{\theta_{\text{old}}}^{(i,t)}}, \qquad g_{i,t}(\theta) = 1. \tag{34}$$

Substituting these into $J_{\text{off}}$ shows that the corrected teacher-output term reduces to the same ratio form as standard GRPO. In other words, once a teacher token has gained sufficient support under $\pi_{\theta_{\text{old}}}$, OnePO no longer applies any low-probability-specific amplification, and the update follows the standard clipped policy-optimization form.

# E. Details of Teacher Retirement

**Setup.** For a fixed prompt $q$, let $\{o_{\text{on}}^{(i)}\}_{i=1}^{G} \sim \pi_{\theta_{\text{old}}}(\cdot \mid q)$ be the $G$ on-policy outputs, with rewards $r_{\text{on}}^{(i)} \triangleq R(o_{\text{on}}^{(i)})$. We also have $K$ teacher outputs $\{o_{\text{off}}^{(k)}\}_{k=1}^{K}$ with rewards $r_{\text{off}}^{(k)} \triangleq R(o_{\text{off}}^{(k)})$.

The retirement rule determines a retained subset $S \subseteq \{1, \ldots, K\}$ of teacher outputs, and we write $n \triangleq |S|$ for the realized number of retained outputs. Let $\{\tilde{r}_{\text{off}}^{(t)}\}_{t=1}^{n}$ denote the retained teacher-output rewards indexed by $S$. To keep the group size fixed at $G$, OnePO replaces a uniformly random size-$n$ subset $I \subseteq \{1, \ldots, G\}$ of on-policy positions, and we denote the remaining positions by $J = \{1, \ldots, G\} \setminus I$.

The resulting mixed reward multiset is

$$\mathcal{R}_{\text{mix}} = \{r_{\text{on}}^{(j)} : j \in J\} \uplus \{\tilde{r}_{\text{off}}^{(t)}\}_{t=1}^{n}.$$

For the distributional equivalence result below, we assume that the on-policy rewards $r_{\text{on}}^{(1)}, \ldots, r_{\text{on}}^{(G)}$ are i.i.d. The teacher-output rewards are treated as fixed once the retained subset $S$ is realized.

## E.1. Algorithm of Teacher Retirement

The detailed procedure is shown in Algorithm 1.

---

**Algorithm 1** Teacher Retirement (per prompt $q$)

---

**Require:** prompt $q$, group size $G$, number of teacher outputs $K$
1: Sample $\{o_i^{\text{on}}\}_{i=1}^{G} \sim \pi_{\theta_{\text{old}}}(\cdot \mid q)$
2: Retrieve $\{o_j^{\text{off}}\}_{j=1}^{K}$ (pre-generated from $\pi_\phi$)
3: Compute $\{r_i^{\text{on}}\}_{i=1}^{G}$ and $\{r_j^{\text{off}}\}_{j=1}^{K}$ via reward $R(\cdot)$
4: $T \leftarrow \max_i r_i^{\text{on}}$
5: **for** $j = 1$ to $K$ **do**
6:     **if** $r_j^{\text{off}} > T$ **then**
7:         Sample an index $u \sim \text{Unif}(\{1, \ldots, G\})$
8:         Replace $o_u^{\text{on}} \leftarrow o_j^{\text{off}}$
9:     **else**
10:         **retire** $o_j^{\text{off}}$
11:     **end if**
12: **end for**
13: **return** a mixed group of size $G$ for OnePO optimization

---

*Remark.* The algorithm is written for a general teacher-output batch size $K$. In all experiments reported in the main text, we use $K = 1$, i.e., one pre-generated teacher output per prompt.

## E.2. Replacement View

Conditioned on a realized retained subset $S$, the replacement step can be viewed as first fixing the retained teacher-output rewards and then uniformly choosing which on-policy positions are replaced. The following proposition shows that, under this view, the remaining on-policy rewards still follow the same law as $G - n$ i.i.d. on-policy draws.

**Proposition E.1** (Random replacement preserves the law of retained on-policy rewards)**.** *Conditioned on a realized retained subset $S$ with $|S| = n$, let $I \subseteq \{1, \ldots, G\}$ be sampled uniformly over all subsets of size $n$, and let $J = \{1, \ldots, G\} \setminus I$. If the on-policy rewards $r_{\text{on}}^{(1)}, \ldots, r_{\text{on}}^{(G)}$ are i.i.d., then the unordered multiset*

$$\{r_{\text{on}}^{(j)} : j \in J\}$$

*has the same distribution as $G - n$ i.i.d. draws from the on-policy reward distribution. Equivalently, for any permutation-invariant function $\phi$,*

$$\mathbb{E}\left[\phi\left(\{r_{\text{on}}^{(j)} : j \in J\}\right) \,\Big|\, S\right] = \mathbb{E}\left[\phi\left(r_{\text{on}}^{(1)}, \ldots, r_{\text{on}}^{(G-n)}\right)\right].$$

*Proof.* Because the on-policy rewards are i.i.d., their joint law is exchangeable. After conditioning on the realized retained subset $S$ and hence on $n = |S|$, the retained on-policy index set $J$ is uniformly distributed over all subsets of size $G - n$. Therefore, the unordered multiset indexed by $J$ has the same distribution as any fixed collection of $G - n$ coordinates, for example $\{r_{\text{on}}^{(1)}, \ldots, r_{\text{on}}^{(G-n)}\}$. $\qquad\square$

## F. Dataset Construction

### F.1. Prompt for Open-Ended Question and Rubric Generation

We use the following fixed instruction template, and call **GPT-5-Chat** to convert the filtered PMC-OA case reports into prompt and rubric sets. The model is required to output strict JSON, the dialogue contains user turns, and no more than 8 rubric items are generated.

---

**Rubric Generation**

```
<Reference prompt>

{
    "role": "user",
    "content": "What is the reason for a persistent ulcer on my forearm. I was
        traveling in the tropics recently and I'm worried it could be leishmaniasis"
},
{
    "role": "assistant",
    "content": "A persistent skin ulcer that appears after travel to tropical regions
        can have several possible causes:\

1. **Cutaneous leishmaniasis**: A parasitic infection transmitted by sand flies.\
2. **Fungal infections**: Certain fungi can create ulcers or sores, especially in
    warm, humid places.\
3. **Bacterial infections**: Minor cuts or bites can become infected and develop into
    non-healing ulcers.\

It's important to see a healthcare professional to confirm the diagnosis and rule out
    other causes."
},
{
    "role": "user",
    "content": "What is the recommended treatment if it's cutaneous leishmaniasis. I'
        m not sure if we have specialized labs or advanced medication available. I'd
        also like to know any home remedies"
}

</Reference prompt>

<Reference rubrics>

[
  {
    "criterion": "States that the recommended treatment for cutaneous leishmaniasis
        includes systemic therapies such as oral miltefosine, pentavalent antimonials
        like sodium stibogluconate, and amphotericin B, as well as local physical
        therapies like cryotherapy, paromomycin ointment, and intralesional
        antimonials.",
    "points": 8
  },
  {
    "criterion": "States that the treatment of cutaneous leishmaniasis depends on
        various factors, including the species of Leishmania, geographic location,
        severity of the lesions, and associated visceral involvement.",
```

---

```
      "points": 7
  },
  {
    "criterion": "States that it is crucial to seek urgent medical care if the ulcer
        persists for 2-3 weeks, if there are multiple lesions, or if the lesions
        involve sensitive areas like the eyes, mouth, or mucous membranes. Signs of
        spreading infection, such as redness, swelling, or fever, also warrant
        immediate medical attention.",
    "points": 6
  },
  {
    "criterion": "Mentions that confirming the diagnosis of cutaneous leishmaniasis
        is crucial, as treatment can be long-term, and that diagnostic methods may
        include direct microscopy, clinical examination, culture, and PCR testing to
        confirm the presence of the parasite. Therefore, even in areas with limited
        access to specialized labs or advanced medications, medical consultation and
        direct microscopic examination can be valuable in establishing a diagnosis and
         guiding treatment.",
    "points": -8
  },
  {
    "criterion": "Provides guidance to seek immediate medical care, emphasizing that
        such services are usually available in endemic areas and that delays in
        treatment can lead to complications.",
    "points": 8
  },
  {
    "criterion": "Response uses natural language and correct interpretation: Response
         uses correct and natural language for the user's language or region. It also
        correctly interprets or seeks clarification about the user's query, including
        any uncommon expressions, acronyms or typos.",
    "points": 5
  },
  {
    "criterion": "Does not mention that some home remedies, including garlic or
        turmeric-based remedies and antihistamines, are not curative for cutaneous
        leishmaniasis. While these may provide temporary symptom relief, they do not
        treat the underlying disease.",
    "points": -9
  },
  {
    "criterion": "States that supportive measures can be taken at home to reduce
        symptoms, such as keeping the ulcer clean and dressed with breathable
        dressings, and avoiding scratching or picking at the lesion. However, these
        measures cannot cure cutaneous leishmaniasis.",
    "points": 5
  }
]

</Reference rubrics>

The text above shows a sample medical evaluation task:

* <Reference prompt> is the question(s) posed to the LLM (possibly multi-turn).

* <Reference rubrics> are the scoring rules (including possible penalties or rewards)
    .
  A higher score means better LLM performance.

Your task:
1. Based on the <Case Report> below, create a medical prompt. Your prompt should
    ideally be challenging and diverse.
```

```
2. Design corresponding rubrics that fairly evaluate the LLM's response quality,
     tailored to the details of the case. You can refer to the <Reference rubrics> to
     ensure your rubrics are of comparable quality and comprehensive. The rubrics
     should have no more than 8 items.

<Case Report>

<<Case Report>>

</Case Report>

Output strictly in the following JSON format:

'''json
{
  "prompt": [
    {"role": "user", "content": "Question"}
  ],
  "rubrics": [
    {"criterion": "Specific guideline", "points": 10},
    {"criterion": "Specific guideline", "points": -8}
  ]
}
```

## F.2. Data Examples

Here we give representative examples from the 20K dataset: one closed-ended multiple-choice sample from HuatuoGPT-o1, and one open-ended sample converted from a PMC-OA case report together with its rubric.

**Closed-ended Example (Multiple-Choice).** The following example shows the format in the verifiable multiple-choice subset. The model is required to output only the option, and place the final answer within the <answer> tag.

---

**Closed-ended example**

```
User:
Answer the multiple-choice question and place the final selection within the <answer>
    tag. Example: <answer>A</answer>.
The person whose work on the radical cure of hernia immortalised his name was -
A. William Halsted
B. Eduardo Bassini
C. Mc Vay
D. Koontz

Reference answer: <answer>B</answer> (Eduardo Bassini)
```

---

**Open-ended Example (Case Report + Rubric).** The open-ended subset comes from PMC-OA case reports. Each case is converted into a challenging task prompt (possibly multi-turn), and is accompanied by a rubric set of no more than 8 items, used to positively and negatively score clinically relevant criteria.

---

**Open-ended example**

```
Prompt (single-turn):
Patient: 28-year-old female, housewife
Chief concerns: Two-year history of numbness, tingling, and slowly progressive
    painful swelling on the medial aspect of the left upper arm; pain has worsened
```

```
        over the past few weeks. Symptoms can be provoked by prolonged wrist flexion/
        extension.
Exam: Palpable pulsatile mass in the left upper arm.
Imaging: Duplex Doppler confirms a brachial artery aneurysm. CT angiography with
        iodinated contrast was deferred due to renal impairment.
History: No prior medical or surgical interventions; no history of trauma, iatrogenic
         puncture, arteriovenous fistula, infection, connective tissue disorder, or drug
        misuse. Duplex vein mapping shows small upper limb veins; both great saphenous
        veins are small and not suitable for harvest.
Vitals: Within normal limits during admission.

Task: Write a formal referral letter to a vascular surgeon for evaluation and
        management of a symptomatic left brachial artery aneurysm. The letter must:
(1) Begin exactly with: "I am referring Ms. [Patient], a 28-year-old female with a
        symptomatic left brachial artery aneurysm, for evaluation and consideration of
        surgical repair."
(2) Address the recipient as Dr. [Surgeon's Surname] ...
...
Do not add unsupported history ... or recommend imaging strategies that conflict with
         the renal impairment.

Rubrics (excerpt, up to 8 items total):
+ Starts the letter exactly with the required opening sentence. (+10)
+ Addresses the recipient as Dr. [Surgeon's Surname]. (+4)
+ Summarizes presentation/exam accurately (2-year numbness/tingling; painful medial
        swelling; palpable pulsatile mass). (+8)
+ Emphasizes renal impairment (elevated creatinine/urea) and justifies deferring
        iodinated contrast CTA; recognizes duplex as primary imaging. (+10)
+ Recommends open aneurysm resection with interposition graft; contrasts endovascular
         stent grafting with pros/cons. (+10)
+ Justifies graft material choice (e.g., ePTFE) due to small veins and need to
        preserve veins for possible future dialysis access. (+8)
+ Perioperative + postoperative plan: heparinization, anastomosis, pulse checks;
        antiplatelet + duplex surveillance (~1 month to 6 months). (+8)
- Penalty: introduces unsupported history or contradicts renal impairment /
        inappropriate management. (-10)
```

## G. Details of Dataset and Benchmarks

**Training Data Construction.**    We construct a 20K training dataset that contains 10K **closed-ended** and 10K **open-ended** medical tasks.

*Closed-ended (10K).* Following HuatuoGPT-o1 (Chen et al., 2024a), we sample 10K high-difficulty medical multiple-choice questions from the training sets of MedQA and MedMCQA. Such tasks have objectively verifiable standard answers, which is convenient for constructing stable verifiable reward.

*Open-ended (10K).* We select case report articles from PMC-OA to construct 10K open-ended tasks (Lin et al., 2023). For each case, we use GPT-5-Chat to generate a challenging prompt and the corresponding rubric (OpenAI, 2025) (the prompt is shown in Appendix F.1). To ensure rubric quality, we add a **verification filtering** step: GPT-5-Chat checks the generated rubric back against the original case text and removes samples that contain hallucinations or clinically unreasonable criteria. This process can be scaled to larger corpora with limited manual annotation, followed by targeted quality checks.

Representative examples are shown in Appendix F.2.

**Benchmarks.**    We evaluate domain-specific capabilities on four medical benchmarks, and additionally report IFEval and GSM8K to observe general capabilities. **HealthBench** evaluates the quality of open-ended health responses using rubric-based grading (Arora et al., 2025); we report the official total score as well as the *Hard* subset. For closed-ended evaluation, we use **Medbullets (5-option)**(Chen et al., 2025a), the **medical subset of MMLU-Pro**(Wang et al., 2024), and **MedXpertQA (Text)** (Zuo et al., 2025), which respectively cover exam-style medical knowledge reasoning, higher-difficulty

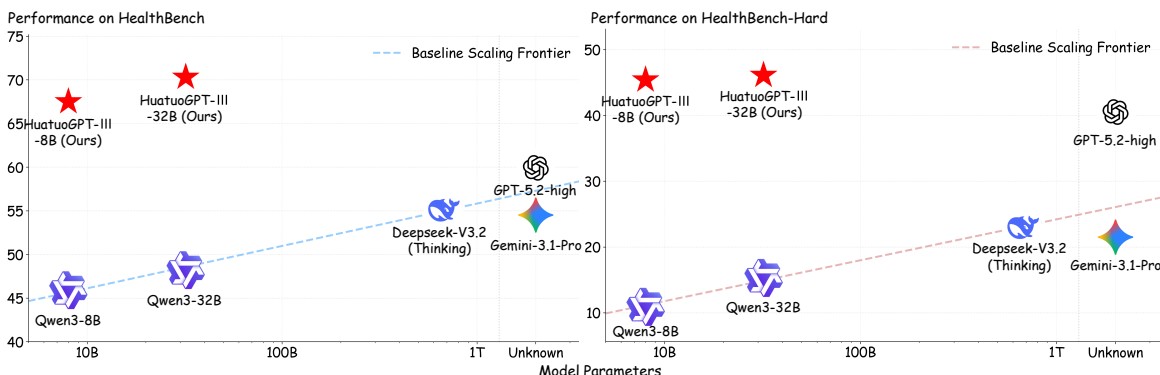

*Figure 8.* HealthBench and HealthBench-Hard scaling frontier. HuatuoGPT-3-8B achieves stronger performance with fewer parameters, and lies above the envelope curve.

comprehensive medical reasoning, and expert-level textual medical question answering. In addition, we include **IFEval** and **GSM8K** as general capability tests (Zhou et al., 2023; Cobbe et al., 2021), which are respectively used to observe whether instruction-following and mathematical reasoning remain stable. IFEval and GSM8K are evaluated zero-shot on their official evaluation sets and are not used for training. Closed-ended benchmarks use exact-match accuracy (%), while HealthBench uses the official evaluation code with GPT-4.1 as the judge model. Figure 8 further shows the position of our model on the HealthBench scaling frontier relative to compared reference models.

## H. Ablation of Reward Data Composition

We perform ablation on the reward composition in the training data to analyze the individual and joint effects of verifiable reward and rubric-based reward. All experiments use Qwen3-8B-Base as the base model, GPT-5-Chat as the teacher source, and OnePO as the training paradigm.

*Table 6.* Ablation of training data composition. **Verifiable Only**: use only 10K closed-ended multiple-choice data and verifiable reward. **Rubric Only**: use only 10K open-ended clinical tasks and rubric-based reward. **Mixed (Ours)**: use the full 20K data and jointly use the two rewards. The mixed setting achieves the best overall trade-off performance, indicating that dense rubric supervision and sparse verifiable signal are complementary.

| | Open-ended | | Closed-ended | | |
|---|---|---|---|---|---|
| Setting | HealthBench (Total) | HealthBench (Hard) | Medbullets | MMLU-Pro | MedXpertQA |
| Qwen3-8B-Base | 22.1 | 0.0 | 30.7 | 41.2 | 11.7 |
| w/ Verifiable Only | 24.2 | 3.2 | **64.5** | 80.5 | 24.8 |
| w/ Rubric Only | 63.7 | 36.6 | 44.8 | 67.0 | 12.5 |
| w/ Mixed (Ours) | **65.4** | **39.1** | 64.0 | **81.2** | **24.9** |

**Analysis.** The results show a clear division of specialization:

- **Verifiable reward only** performs strongly on closed-ended benchmarks (e.g., Medbullets 64.5%, MMLU-Pro 80.5%), but can hardly improve open-ended generation quality (HealthBench Total 24.2%, only slightly higher than the 22.1% of the base model). This shows that although verifiable reward provides explicit optimization signals for factual knowledge, it is difficult to transfer to open-ended tasks that require fine-grained clinical expression and reasoning.

- **Rubric reward only** is effective on open-ended evaluation (e.g., HealthBench Total 63.7%, Hard 36.6%), but clearly drops on closed-ended tasks (e.g., Medbullets 44.8%, significantly lower than verifiable-only). This shows that dense rubric supervision can effectively shape open-ended response quality, but may also over-constrain reasoning processes, thereby weakening task performance where concise and definite answers are required.

- **Mixed reward (our setting)** balances the two types of tasks at the same time: it not only maintains strong open-ended

*Table 7.* Ablation results of different training settings (Pure RL / SFT / SFT+RL / OnePO) under different teacher sources.

| Teacher Source | Setting | Open-ended (HealthBench) | | Closed-ended | | |
| | | Total | Hard | Medbullets | MMLU-Pro(Med) | MedXpertQA |
| --- | --- | --- | --- | --- | --- | --- |
| None | Pure RL | 59.8 | 25.2 | 48.1 | 75.0 | 20.0 |
| GPT-5-Chat | SFT | 30.4 | 4.8 | 49.7 | 72.5 | 21.7 |
| | SFT+RL | 63.6 | 37.3 | 61.9 | 78.2 | 21.5 |
| | OnePO | 65.4 | 39.1 | 64.0 | 81.2 | 24.9 |
| DeepSeek-V3.2-Thinking | SFT | 36.8 | 2.4 | 56.6 | 71.0 | 17.2 |
| | SFT+RL | 64.5 | 40.7 | 63.5 | 80.0 | 22.5 |
| | OnePO | 67.2 | 44.5 | 65.2 | 82.0 | 25.9 |

performance (HealthBench Total 65.4%), but also preserves strong closed-ended capability (Medbullets 64.0%, MMLU-Pro 81.2%). This shows that the two reward types are clearly complementary: rubric reward provides controllable and dense behavior guidance, while verifiable reward helps preserve freer reasoning space.

## I. SFT Baseline on HealthBench

Table 7 shows that SFT alone obtains a relatively low HealthBench score compared with Pure RL. This does not indicate that the teacher data are harmful: with the same GPT-5-Chat teacher source, SFT+RL improves over Pure RL on HealthBench (63.6 vs. 59.8) and closed-ended benchmarks. The issue is that HealthBench evaluates held-out open-ended clinical response quality with rubric-based scoring, rather than imitation fidelity to teacher-style answers. In this setting, naive SFT can learn the surface style of teacher outputs while generalizing poorly to unseen rubric criteria. OnePO uses the same teacher outputs inside reward-driven optimization, which help explain why it outperforms the SFT+RL pipeline under the same data and teacher source.

## J. Sensitivity to Probability Floor

The probability floor $c$ controls how aggressively OnePO promotes low-probability teacher-output tokens. Table 8 sweeps $c$ while keeping other settings unchanged. Setting $c = 0$ removes the floor and causes a large performance drop. Very small values provide insufficient correction for low-probability tokens, while overly large values keep more teacher tokens in the corrected regime and increase the risk of anchoring. We use $c = 0.1$ as the default because it gives the best overall trade-off.

*Table 8.* Ablation of the probability floor $c$. All runs use Qwen3-8B-Base, GPT-5-Chat as the teacher source, and the same OnePO training setup.

| Floor $c$ | HealthBench | Medbullets | MMLU-Pro (Med) |
| --- | --- | --- | --- |
| 0.00 | 49.1 | 49.7 | 73.1 |
| 0.01 | 58.3 | 59.7 | 77.2 |
| 0.05 | 64.8 | 63.2 | 79.5 |
| 0.10 | **65.4** | **64.0** | **81.2** |
| 0.20 | 62.1 | 61.5 | 79.8 |
| 0.50 | 60.2 | 59.1 | 77.3 |

## K. Effect of KL Regularization in the AHA-Medicine Pilot

Our main experiments follow DAPO-style GRPO training and omit additional KL loss or KL reward penalty. To examine whether OnePO's off-policy absorption still works under KL constraints, we extend the AHA-Medicine pilot study from Appendix C.1. We test two common forms of KL regularization: adding a KL term to the objective, and adding a trajectory-level KL penalty to the reward. KL regularization keeps the policy closer to a reference policy, but it does not directly solve the low-probability token learnability problem addressed by the probability floor and gradient rescaling.

Table 9 reports the number of RL steps required to absorb the synthetic AHA-Medicine knowledge, measured by reaching

reward $\geq 0.5$. OnePO still absorbs the off-policy knowledge under KL constraints, while removing either the probability floor or the rescaling term fails within 100 steps.

*Table 9.* KL-constrained AHA-Medicine pilot. "Fail" means the model does not absorb the target knowledge within 100 RL steps. The KL loss coefficient is 0.001, and the KL reward-penalty coefficient is 0.01.

| Method | Steps to reward $\geq 0.5$ |
|---|---|
| OnePO | 7 |
| OnePO w/ KL penalty | 7 |
| OnePO w/ KL loss | 8 |
| OnePO w/ KL loss + KL penalty | 8 |
| OnePO w/o probability floor | Fail |
| OnePO w/o rescaling | Fail |

We further track the log-probability assigned to the target knowledge span "AHA-Medicine" in the teacher output:

$$\frac{1}{t_e - t_s + 1} \sum_{t=t_s}^{t_e} \log \pi_\theta(\tau_t \mid q, \tau_{<t}), \tag{35}$$

where $[t_s, t_e]$ is the token span corresponding to "AHA-Medicine." Values closer to 0 indicate stronger absorption of the off-policy knowledge. Table 10 shows that KL constraints make updates more conservative but do not remove OnePO's effect, while removing rescaling leaves the target span at very low probability.

*Table 10.* Average log-probability of the target "AHA-Medicine" span across the first eight training steps.

| Method | S1 | S2 | S3 | S4 | S5 | S6 | S7 | S8 |
|---|---|---|---|---|---|---|---|---|
| OnePO | -6.9735 | -6.1488 | -4.2228 | -2.7612 | -1.0261 | -0.2838 | -0.0156 | -0.0122 |
| OnePO w/ KL loss + KL penalty | -6.9735 | -6.6096 | -5.5103 | -3.5550 | -2.2576 | -1.0866 | -0.3733 | -0.0406 |
| OnePO w/o rescaling | -6.9735 | -7.1113 | -7.1156 | -7.1270 | -6.9345 | -6.9741 | -6.5402 | -6.6415 |

Table 11 reports the rollout-based empirical KL estimate used during training. KL constraints reduce policy movement as expected. This supports the intended interpretation: KL acts as a regularizer on policy movement, whereas OnePO improves the learnability of low-probability off-policy tokens inside the RL objective.

*Table 11.* Rollout-based empirical KL estimates in the AHA-Medicine pilot. Small negative values may appear due to sampling noise.

| Method | S1 | S2 | S3 | S4 | S5 | S6 | S7 | S8 | Avg. |
|---|---|---|---|---|---|---|---|---|---|
| OnePO | 0.0000 | 0.0645 | 0.0608 | 0.0280 | 0.0038 | 0.0047 | 0.0072 | 0.0064 | 0.0232 |
| OnePO w/ KL loss + KL penalty | 0.0000 | 0.0410 | 0.0238 | 0.0072 | -0.0125 | 0.0015 | 0.0033 | 0.0018 | 0.0091 |
| OnePO w/o rescaling | 0.0000 | 0.0533 | 0.0590 | 0.0755 | 0.1160 | 0.1456 | 0.1471 | 0.1779 | 0.0961 |

## L. Cross-Domain Validation

To test whether OnePO is tied to medicine, we evaluate it on two additional domains with a different base model, Qwen3-4B-Base. The writing task uses 10K samples from the writing subset of RubricHub-v1 with rubric-based rewards, and the legal task uses 10K multiple-choice questions from Unilaw-R1-Data with exact-match rewards. GPT-5-Chat is used as the teacher source in both domains. These two settings cover both open-ended rubric-based optimization and closed-ended verifiable optimization outside medicine.

## M. Rubric Filtering and Grader Validation

For open-ended medical training data, each generated rubric is checked against the source case report and the generated prompt. A rubric item is removed if it is unsupported by the case report, contradictory to the case facts, overly generic or unnecessary, clinically unreasonable or unsafe, or inappropriate for the question. Each rubric is checked twice by LLMs; if

*Table 12.* Cross-domain validation on writing and law using Qwen3-4B-Base. OnePO improves over SFT+RL under the same teacher source in both domains.

| Method | Writing | | Law | |
|---|---|---|---|---|
| | **WritingBench** | **CreativeWriting-v3** | **LexEval** | **LawBench** |
| Qwen3-4B-Base | 33.7 | 23.9 | 18.6 | 41.8 |
| w/ Pure RL | 67.4 | 41.1 | 46.5 | 60.8 |
| w/ SFT+RL | 71.2 | 45.6 | 50.7 | 64.5 |
| w/ OnePO | **74.6** | **47.8** | **51.9** | **67.4** |

either check flags a problem, the item is removed. If three or more rubric items are removed from a sample, we discard the entire sample. Under this filtering process, 11.6% of rubric items and 7.4% of training samples are filtered out.

We additionally sampled 130 question-rubric pairs and asked a senior medical student to check whether each rubric meets the quality standard, with access to the source case report and web search when needed. The human validation failure rate is 3.1% (4/130), suggesting that the filtered rubrics are reasonably reliable.

To evaluate grader reliability, we sampled 200 query-answer-rubric-criterion instances from OnePO outputs on our open-ended data. Each instance was annotated by a doctor, and we compare both the training-time 8B grader and the official GPT-4.1 evaluation grader against the human label. As shown in Table 13, GPT-4.1 has stronger agreement with human judgment, while the 8B grader remains reasonably aligned for training-time reward estimation. Importantly, HealthBench results in the paper are evaluated with the official GPT-4.1 grader, not the 8B training-time grader.

*Table 13.* Agreement between graders and doctor annotations on 200 query-answer-rubric-criterion instances.

| Metric vs. Human Label | 8B Grader (Training) | GPT-4.1 Grader (Evaluation) |
|---|---|---|
| F1 score | 0.81 | **0.85** |
| Cohen's kappa | 0.62 | **0.72** |

We also re-evaluate the same model outputs with different graders. Table 14 shows that absolute scores vary across graders, but the gain from OnePO remains consistent. This does not fully rule out reward-model bias, but it reduces the concern that the improvement is specific to a single grader.

*Table 14.* HealthBench re-evaluation with different graders.

| Model | 8B Grader | GPT-4.1 Official | GPT-5-Chat Grader |
|---|---|---|---|
| Qwen3-8B | 48.5 | 45.9 | 41.8 |
| Qwen3-8B w/ OnePO | 70.7 | 67.2 | 65.1 |
| Gain | +22.2 | +21.3 | +23.3 |

## N. Reward Stability

*Table 15.* Training reward curves for open-ended rubric-based rewards and closed-ended verifiable rewards.

| Train step | 0 | 20 | 40 | 60 | 80 | 100 | 120 |
|---|---|---|---|---|---|---|---|
| Open-ended reward | 0.34 | 0.69 | 0.69 | 0.70 | 0.70 | 0.75 | 0.77 |
| Closed-ended reward | 0.29 | 0.38 | 0.48 | 0.52 | 0.55 | 0.58 | 0.59 |

Rubric-based rewards are instance-specific but fixed during training: each prompt has a fixed rubric, and the reward model evaluates candidate responses against that rubric. Table 15 shows the training reward curves for open-ended and closed-ended subsets. Open-ended reward improves smoothly, while closed-ended reward also increases steadily under exact-match supervision. This suggests that combining dense rubric-based rewards with verifiable rewards provides stable optimization signals in our setting.

