# OpenReview forum: "OnePO: Direct One-stage Policy Optimization for SFT-free Domain Adaptation"
_ICML.cc/2026/Conference — ICML 2026 regular_

### Official Review · Reviewer_9rQA · 2026-03-11

**Soundness:** 3
**Presentation:** 3
**Significance:** 2
**Originality:** 2
**Overall Recommendation:** 4
**Confidence:** 3

**Summary:**

This paper proposes OnePO, a one-stage reinforcement learning framework for domain adaptation that eliminates the need for supervised fine-tuning as a cold-start phase. The central argument is that pre-SFT is fundamentally problematic: it blindly clones reference distributions, contracts the output space, and burdens the training pipeline. Pure RL and existing mixed-policy methods both fail to fill this gap—pure RL cannot acquire off-policy knowledge, and mixed-policy RL suffers from gradient starvation or anchoring. OnePO addresses this with two mechanisms. Adaptive Objective Evolution introduces a probability floor and a reweighting term that recovers SFT-level learning efficiency for off-policy tokens in early training, then automatically transitions to standard GRPO as the model matures. Teacher Retirement discards off-policy trajectories once the on-policy model surpasses them, preventing the model from converging toward the teacher's distribution. Applied to medical domain adaptation with only 20K samples, OnePO trains Qwen3-8B-Base into a competitive medical LLM, achieving 67.2 on HealthBench and outperforming SFT+RL on multiple benchmarks.

**Compliance With Llm Reviewing Policy:**

Affirmed.

**Key Questions For Authors:**

**Strengths**

The gradient starvation analysis in Section 2.1 is technically precise and explains why existing mixed-policy methods struggle with off-policy tokens. The AHA-Medicine pilot study is a well-designed non-leakage test that separates knowledge acquisition from exploration. The two mechanisms are simple and do not require substantial additional hyperparameter tuning. Table 4 provides useful evidence that OnePO avoids the capability regression associated with SFT.

**Weaknesses**

1. The paper validates on one domain with one base model. The central claim is that OnePO is a general SFT-free paradigm for domain adaptation, but this is never tested beyond medical. This substantially weakens the paper's generalization argument.

2. The SFT baseline in Table 6 scores only 30.4 on HealthBench with GPT-5-Chat as teacher, far below pure RL at 59.8. This is never explained. If the SFT data is problematic for HealthBench evaluation, then the SFT+RL comparison may be skewed in favor of OnePO, which undermines the main empirical claim.

3. The probability floor c is the most important hyperparameter in the method—it controls when and how long the SFT-like regime operates. No ablation over c is provided.

4. The rubric reward pipeline relies entirely on GPT-5-Chat for generation and GPT-4.1 for scoring, with no human validation. Biases in this pipeline could distort results for open-ended tasks, which make up half the training data.

**Limitations:**

1. What happens to performance when c is set to 0.05 or 0.2? Given that c controls the core learning dynamic, the absence of this ablation is a gap.
2. Has the rubric generation pipeline been checked against human expert judgment? What fraction of rubrics were discarded during filtering?
3. Is there any evidence that OnePO works outside the medical domain?

**Strengths And Weaknesses:**

This paper proposes OnePO, a one-stage RL framework for domain adaptation that removes the SFT cold-start phase. The argument is that pre-SFT causes distribution contraction and adds unnecessary overhead. Two mechanisms are introduced: Adaptive Objective Evolution recovers SFT-level learning efficiency for off-policy tokens and transitions automatically to standard GRPO, and Teacher Retirement discards off-policy trajectories once the on-policy model surpasses them. Experiments are conducted in the medical domain using 20K samples, training Qwen3-8B-Base to 67.2 on HealthBench and outperforming SFT+RL baselines.

---

> ### Author Rebuttal · Authors · 2026-03-31
>
> We sincerely appreciate the reviewer's thoughtful evaluation and valuable feedback. We respond to each concern below:
>
> > **W1 & Q3. The paper only validates one domain and one base model.**.
>
> We agree and extend our experiments with two new domains and a new backbone:
>
> - **New domains.** We include two new domains: *writing (open-ended, rubric-based reward)* and *law (close-ended, verifiable reward)*.
> - **New backbone.** These experiments are conducted on a new backbone `Qwen3-4B-Base`, with `GPT-5-Chat` as the off-policy teacher.
>
> ### ✍️ $\color{blue}{\text{Writing domain (Open-End)}}$
>
> We use 10K writing samples from the writing subset of *RubricHub-v1* as training data. Its prompt-rubric format aligns naturally with our reward setting. Results on *WritingBench* and *Creative Writing v3* are as follows:
>
> | Method | WritingBench | CreativeWriting-v3 |
> |---|-:|-:|
> | Qwen3-4B-Base | 33.7 | 23.9 |
> | w/ Pure RL | 67.4 | 41.1 |
> | w/ SFT + RL | 71.2 | 45.6 |
> | **w/ OnePO** | **74.6** `(+3.4)` | **47.8** `(+2.2)` |
>
> ### ⚖️ $\color{blue}{\text{Law domain (Close-End)}}$
>
> We also train on 10k legal multiple-choice questions from *Unilaw-R1-Data*, using Exact Match as the reward signal. Legal capabilities were evaluated on *LawBench* and *LexEval*:
>
> | Method | LexEval | LawBench |
> |---|---:|---:|
> | Qwen3-4B-Base | 18.6 | 41.8 |
> | w/ Pure RL | 46.5 | 60.8 |
> | w/ SFT + RL | 50.7 | 64.5 |
> | **w/ OnePO** | **51.9** `(+1.2)` | **67.4** `(+2.9)` |
>
> **Conclusion:**
> These results provide evidence that OnePO is effective beyond the medical domain and on a different base model.
>
> We thank the reviewer for this suggestion and will include full details in the revision.
>
> > **W2. SFT baseline scores only 30.4 on HealthBench — unexplained.**
>
> Thank you for raising this. We believe the key reason is that *HealthBench evaluates open-ended clinical response quality using rubric-based scoring, rather than simple imitation fidelity*. In this setting, naive SFT can fit teacher-style responses but generalize poorly to held-out rubric-based evaluation.
>
> **This is also consistent with recent external evidence:**
> > `ClinAlign` (Paper source: https://arxiv.org/abs/2602.09653) reports that naive SFT on HealthBench quickly saturates but shows limited held-out gains.
>
> Although *SFT alone* is limited, it still benefits subsequent RL: **63.6 (SFT+RL) vs. 59.8 (Pure RL)**. This suggests that the issue is not the SFT data itself, but rather how it is used. OnePO uses **the same off-policy data** more effectively during RL, which is why it outperforms the traditional SFT+RL pipeline.
>
> > **W3 & Q1. No ablation over probability floor $c$.**
>
> We agree and thank the reviewer for pointing this out. we will add the following ablation study on the probability floor `c`:
>
> | $c$ value | HealthBench | Medbullets | MMLU-Pro (Med) |
> |---|---|---|---|
> | 0.00 | 49.1 | 49.7 | 73.1 |
> | 0.01 | 58.3 | 59.7 | 77.2 |
> | 0.05 | 64.8 | 63.2 | 79.5 |
> | **0.1 (default)** | **65.4** | **64.0** | **81.2** |
> | 0.2 | 62.1 | 61.5 | 79.8 |
> | 0.5 | 60.2 | 59.1 | 77.3 |
>
> The results highlight the importance of **Adaptive Objective Evolution** in OnePO. `c` controls how aggressively low-probability off-policy tokens are promoted: too small is ineffective, while too large increases the risk of over-anchoring. `c = 0.1` gives the best balance.
>
> > **W4 & Q2. Rubric pipeline lacks human validation and filtering transparency.**
>
> We agree that the original submission did not make this part sufficiently transparent. We therefore add the following analyses.
>
> **1. Filtering transparency.**
>
> Each rubric is checked against the source case report and the generated prompt. A rubric item is removed if it is:
>
> - not supported by the case report,
> - contradictory to the case facts,
> - overly generic or unnecessary,
> - clinically unreasonable or unsafe,
> - inappropriate for the question.
>
> Each rubric is checked **twice** by LLMs. If either check flags a problem, that rubric item is removed. If three or more rubric items are removed, we discard the entire sample. The filtering statistics are:
>
> |  | Fraction filtered (%) |
> |-|-:|
> | Rubric items | 11.6 |
> | Training samples | 7.4 |
>
> **2. Human validation.**
>
> To further address this concern, We further sampled *130* `(question, rubric)` pairs and asked a senior medical student to check whether each rubric met our quality standard, with access to the source case report and web search when needed:
>
> |  | Fraction failing human validation |
> |-|-:|
> | Human check | 3.1% (4 / 130) |
>
> These results suggest that the filtered rubrics are reasonably reliable.
>
> **3. Further clarification.**
>
> - Our open-ended training data comes from *PMC case reports*, which is distinct from the *HealthBench* evaluation data.
> - The training-time grader ( `8B model`) and the final evaluation grader (`official GPT-4.1`) are different, reducing the risk of simple reward hacking.
> - OnePO also improves *close-ended tasks* with exact-match rewards, where grader bias is not a concern.

---

> > ### Author Rebuttal · Reviewer_9rQA · 2026-04-08
> >
> > Thanks for the response, I will maintain my score.

---

> > > ### Author Response · Authors · 2026-04-08
> > >
> > > We appreciate the reviewer's feedback and are glad that our rebuttal successfully resolved the concerns raised. Thank you again for your valuable suggestions throughout the review process.

---

### Official Review · Reviewer_VsEe · 2026-03-11

**Soundness:** 3
**Presentation:** 4
**Significance:** 3
**Originality:** 3
**Overall Recommendation:** 4
**Confidence:** 3

**Summary:**

This paper questions whether Supervised Fine-Tuning (SFT) is a necessary cold-start stage before Reinforcement Learning (RL) for domain adaptation of LLMs. The authors argue that SFT causes distribution contraction and capability regression, while pure RL and existing mixed-policy RL methods fail to efficiently acquire off-policy knowledge. They propose OnePO (One-stage Policy Optimization), which combines two mechanisms: (1) Adaptive Objective Evolution, which achieves SFT-level learning efficiency for off-policy tokens in early training, automatically transitioning to standard GRPO as the model matures; and (2) Teacher Retirement, which discards teacher trajectories once they are outperformed by the on-policy model's best rollout, preventing anchoring to the teacher distribution. Validated on a medical dataset (OneData-Health-20K), OnePO transforms Qwen3-8B-Base into OneMed-8B, outperforming the SFT+RL pipeline.

**Compliance With Llm Reviewing Policy:**

Affirmed.

**Final Justification:**

I maintain my score

**Key Questions For Authors:**

1. Have you tested OnePO on any domain besides medicine (e.g., legal, financial, code generation)? Even a smaller-scale experiment would significantly strengthen the generality claim.

2. What is the agreement rate between the 8B LLM grader and GPT on held-out data? Is there evidence that the RL training is not reward-hacking the grader?

**Limitations:**

The paper includes an Impact Statement acknowledging that medical LLMs carry risks if they produce inaccurate advice, and appropriately notes the model is a research prototype not intended for clinical deployment. However, the paper's discussion of technical limitations is notably thin.

**Strengths And Weaknesses:**

S1: Rigorous gradient-level motivation: The paper provides a thorough mathematical analysis of why existing approaches fail for off-policy knowledge injection. The identification of gradient starvation in mixed-policy RL versus SFT's natural prioritization of unfamiliar tokens is precise and well-articulated.

S2: Non-leakage pilot study: The experiment is a well-constructed diagnostic: a purely synthetic drug name ensures no data leakage, the two-tier reward cleanly separates knowledge acquisition from autonomous reasoning, and the results provide direct evidence that Pure RL and LUFFY fail while OnePO succeeds. These controlled experiments add substantial credibility beyond benchmark numbers.

S3: Comprehensive ablations with clear signal: Each OnePO component is ablated individually. The retirement threshold ablation convincingly shows that looser thresholds plateau due to anchoring. The teacher quality robustness experiment demonstrates a small gap between strong and weak teachers, which is a strong practical property.

W1: Single-domain validation undermines the generality of the central claim: The paper's core thesis — that SFT is unnecessary for domain adaptation — is validated exclusively in the medical domain on a single base model (Qwen3-8B-Base). The medical domain has specific properties (structured reasoning, checkable facts, established rubrics) that may make it unusually amenable to RL-based learning. Domains like law, finance, or code generation may have different knowledge-acquisition dynamics (e.g., longer dependency chains, different reward signal density). Without at least one additional domain, the claim "one-stage RL suffices for domain adaptation" remains underspecified. This is the paper's most significant limitation.

W2: Potential reward hacking: The open-ended reward uses an 8B LLM grader trained on 200K GPT-4.1 annotations. Since the training reward and evaluation reward share the same underlying judge, there is a risk that OnePO optimizes for features that please the judge rather than genuine clinical quality. The paper does not analyze reward model accuracy, calibration, or agreement with human judgments. A human evaluation study, even small-scale, would substantially strengthen the claims.

---

> ### Author Rebuttal · Authors · 2026-03-31
>
> We sincerely thank the reviewer for the positive assessment and constructive suggestions. We address your concerns below:
>
> > **W1 & Q1. Single-domain validation undermines generality.**
>
> Thank you for this valuable suggestion. Following the reviewer’s advice,  we added small-scale preliminary validation on **two new domains and a new backbone**.
>
> We include two new domains: **writing (open-ended, rubric-based reward)** and **law(close-ended, verifiable reward)**. These experiments are conducted on a new backbone, `Qwen3-4B-Base`, with `GPT-5-Chat` as the off-policy teacher.
>
> - **✍️ $\color{blue}{\text{Writing domain (Open-End)}}$**
>
> We use 10K writing samples from the writing subset of *RubricHub-v1* as training data. Its prompt-rubric format aligns naturally with our reward setting. Results on *WritingBench* and *Creative Writing v3* are as follows:
>
> | Method | WritingBench | CreativeWriting-v3 |
> |---|---:|---:|
> | Qwen3-4B-Base | 33.7 | 23.9 |
> | w/ Pure RL | 67.4 | 41.1 |
> | w/ SFT + RL | 71.2 | 45.6 |
> | **w/ OnePO** | **74.6** `(+3.4)` | **47.8** `(+2.2)` |
>
> - **⚖️ $\color{blue}{\text{Law domain (Close-End)}}$**
>
> We also train on 10k legal multiple-choice questions from *Unilaw-R1-Data*, using Exact Match as the reward signal. Legal capabilities were evaluated on *LawBench* and *LexEval*:
>
> | Method | LexEval | LawBench |
> |---|---:|---:|
> | Qwen3-4B-Base | 18.6 | 41.8 |
> | w/ Pure RL | 46.5 | 60.8 |
> | w/ SFT + RL | 50.7 | 64.5 |
> | **w/ OnePO** | **51.9** `(+1.2)` | **67.4** `(+2.9)` |
>
> - **$\color{blue}{\text{Conclusion}}$**
>
> These additional experiments provide preliminary evidence that **OnePO** remains effective beyond medicine, across diverse reward scenarios.
>
> We thank the reviewer for this constructive feedback. We will include full experimental details in the revised version.
>
>
> > **W2 & Q2. Potential reward hacking: the grader is model-based and agreement is not analyzed.**
>
> Thank you for raising this important concern. We agree that analyzing the reliability of the grader is important, especially because the training reward is model-based.
>
>
> First,  We would like to respectfully clarify that all HealthBench results reported in the paper use the `official GPT-4.1 grader`, not the `8B grader`. The 8B grader is used **only during RL training** for cost efficiency.
> Therefore, the evaluation is separated from the training-time reward model.
>
> Second, following the reviewer’s suggestion, we add two analyses to further address the reward-hacking concern:
>
> - **$\color{blue}{\text{(1) Agreement with human judgment}}$**
>
> We sampled **200** `(query, answer, rubric criterion)` instances from OnePO outputs on our open-ended data.  Each instance was annotated by a doctor, and we computed agreement between each grader and the corresponding human label on that instance.
>
> |  | 8B grader (training) | GPT-4.1 grader (evaluation) |
> |---|---:|---:|
> | F1 score vs. human labels | 0.81 | 0.85 |
> | Agreement with the human label (Cohen’s kappa)  | 0.62 | 0.72 |
>
> **$\color{blue}{\text{Observation.}}$**
> GPT-4.1 shows **substantial agreement with human judgment** (`Cohen’s kappa = 0.72`), while the 8B grader is clearly weaker.  This suggests that the final evaluator used in our paper is better aligned with human judgment than the 8B reward model used during training.
>
> - **$\color{blue}{\text{(2) Robustness across different graders}}$**
>
> To further reduce concern that the gain is specific to one judge, we re-evaluate the same model outputs with different graders.
>
> |  | HealthBench (`8B grader`) | HealthBench (`GPT-4.1, official`) | HealthBench (`GPT-5-Chat grader`) |
> |---|---:|---:|---:|
> | Qwen3-8B | 48.5 | 45.9 | 41.8 |
> | Qwen3-8B w/ OnePO | 70.7 | 67.2 | 65.1 |
> | Gain over Qwen3-8B | **+22.2** | **+21.3** | **+23.3** |
>
> **$\color{blue}{\text{Observation.}}$**
> The absolute scores vary somewhat across graders, and stronger graders tend to assign slightly lower scores. However, the **gain from OnePO remains consistently large under all three graders**. This reduces the concern that the observed gains are specific to a single grader, although it does not fully rule out reward-model bias.
>
> We sincerely thank the reviewer for this valuable suggestion. We will include these analyses more clearly in the revised version.
>
> > **Limitation. Technical limitations are under-discussed.**
>
> Thank you for pointing this out. We agree and will strengthen the limitation discussion in the revised version.
>
> In particular, OnePO still depends on the quality of the off-policy teacher and reward signal and has only been preliminarily tested on a few domains. Furthermore, model-based grading cannot fully eliminate reward misspecification or reward-hacking risks, and we have not explored combining multiple off-policy responses or models.

---

> > ### Author Rebuttal · Reviewer_VsEe · 2026-04-03
> >
> > Maintain 4 (weak accept), with increased confidence. Both concerns are adequately addressed. The cross-domain results strengthen generality, and the grader analysis adds needed rigor. The contribution is solid.

---

> > > ### Author Response · Authors · 2026-04-06
> > >
> > > Thank you for the constructive feedback and the positive recommendation. We’re glad that our rebuttal resolved your concerns. Thanks again!

---

### Official Review · Reviewer_ff1Q · 2026-03-12

**Soundness:** 2
**Presentation:** 3
**Significance:** 3
**Originality:** 3
**Overall Recommendation:** 4
**Confidence:** 4

**Summary:**

The paper presents OnePO, a reinforcement learning framework for domain adaptation of large language models that attempts to eliminate the conventional Supervised Fine-Tuning (SFT) -> RL training pipeline. Instead, the authors propose a single-stage RL approach that incorporates off-policy teacher trajectories directly into the RL objective. The method introduces two mechanisms: Adaptive Objective Evolution, which modifies the off-policy RL objective using a probability floor and reweighting term to improve learning of low-probability tokens, and Teacher Retirement, which removes teacher trajectories once the student model surpasses them in reward.

**Compliance With Llm Reviewing Policy:**

Affirmed.

**Final Justification:**

As presented in the Rebuttal Acknowledgement,  I am still unclear on the KL divergence in PPO; "To clarify, following DAPO, we omit all KL constraints in all our experiments (Section 4.1)." - This is convincing.

However, I am still unclear and not convinced (conceptually) with the experiment presented to back the prior claims on KL divergence.

However, I have raised my score.

**Key Questions For Authors:**

Please see above weakness.

**Limitations:**

Yes

**Strengths And Weaknesses:**

Strengths
1. The paper explores an interesting direction: integrating SFT and reinforcement learning within a single optimization stage.
2. The proposed mechanisms (probability floor, reweighting, and teacher retirement) are conceptually simple and could potentially be integrated into existing RL pipelines.
3. Empirical improvements over the base model are reported across several medical benchmarks. The approach is evaluated in the medical domain consisting of both multiple-choice and open-ended questions. The authors report improvements over pure RL and the conventional SFT + RL pipeline.
4. The paper includes ablation studies that attempt to isolate the contributions of the proposed components.

Weaknesses
1. Weak motivation for bypassing SFT in introduction: The central premise of removing the SFT stage is not convincingly justified. Even when off-policy teacher data is available, the standard and widely adopted pipeline in LLM post-training is SFT on teacher trajectories -> RL for improvement. SFT is typically faster to converge and more sample-efficient than RL. Since the proposed method still relies on teacher trajectories (generated by external models or curated datasets), it is unclear why performing SFT first would be undesirable. The claims that SFT increases complexity, data curation cost, and restricts exploration do not sufficiently justify the need for the proposed method. Moreover, the method assumes the availability of teacher trajectories, which weakens the argument regarding expensive data curation. Overall, the motivation for eliminating SFT is not well established.
2. The paper attributes difficulty in learning off-policy knowledge to the clipping mechanism in PPO-style objectives. However, policy-gradient methods used in LLM RL training (e.g., TRPO, PPO) typically constrain updates using KL divergence. In practice, PPO implementations often include an additional KL penalty term alongside clipping. Under such constraints, it remains unclear how the proposed probability floor and reweighting mechanism would enable the model to learn knowledge that lies far outside the prior distribution.
3. The paper claims that the proposed method performs token-level optimization that is more fine-grained than sequence-level SFT. However, standard SFT with cross-entropy loss already operates at the token level since the loss is computed over individual token predictions. The distinction between the optimization granularity of OnePO and SFT is therefore unclear.
4. The paper states that off-policy trajectories may come from external models or curated datasets and refers to them collectively as the “teacher.” However, the experimental section does not clearly specify whether the teacher responses are generated or curated, how many teacher responses are used per prompt, or how teacher trajectories are sampled during training. These details are important for reproducibility and understanding the role of teacher supervision.
5. The paper labels GPT-5 as a non-thinking model, whereas reasoning models typically generate internal reasoning traces before producing answers. The criteria used to distinguish “thinking” versus “non-thinking” teachers are unclear.
6. Figure 7 shows that the weak teacher retirement ratio quickly rises to approximately 0.75 early in training. This behavior is counterintuitive; if the teacher provides useful guidance during early training, one would expect teacher trajectories to remain active longer. The paper should clarify why the weak teacher is retired so quickly and whether this behavior is intended.
7. Table 4 reports results on benchmarks such as IFEval and GSM8K but does not clearly specify details on the dataset split settings for training and evaluation.
8. The paper uses rubric-based rewards for open-ended medical tasks but does not provide sufficient details on how these rubrics are created. Providing a clear example and explanation of the rubric design process would improve clarity.
9. Reinforcement learning with dynamically changing, instance-specific rubric-based rewards may introduce instability during training. The paper does not discuss how such instability is mitigated. Providing convergence curves for the open-ended RL tasks would help clarify the stability of the training process.

---

> ### Author Rebuttal · Authors · 2026-03-31
>
> We greatly appreciate the reviewer's thorough review and insightful comments. We address each point below.
>
>
> > **W1. Weak motivation for bypassing SFT.**
>
> We acknowledge that the standard *SFT+RL* pipeline is widely used, and that SFT has clear advantages such as simplicity and fast convergence.
>
> In this paper, however, we explore a newer paradigm: SFT-free training for domain adaptation. Recent post-training works (e.g., DeepSeek-R1-Zero, Open-Reasoner-Zero, SimpleRL-Zoo, CurioSFT) increasingly apply RL directly to base models.  Their shared motivation is that pre-SFT may narrow the solution space for RL. Our experiments are consistent with this: Qwen3-8B-Base has an initial entropy of `2.00`, which drops to `0.22` after SFT, indicating reduced generation diversity.
>
> We therefore believe **RL-only training for domain adaptation is a worthwhile direction to investigate**. Following the reviewer’s suggestion, we will revise all SFT-related statements to make this motivation more precise and balanced.
>
>
> > **W2. Mechanism under Clipping + KL constraints.**
>
> We agree that Some RL algorithms like PPO use KL loss and KL penalty to improve stability, which could constrain off-policy trajectory learning. To test their effect on OnePO, we extend the first pilot study (see Appendix B.1) to assess off-policy knowledge absorption:
>
> |Ablation|RL steps to absorb AHA-Medicine (reward > 0.5) ↓|
> |-|-|
> |OnePO |7|
> |w/ KL penalty |7|
> |w/ KL loss |8|
> |w/ KL loss + KL penalty |8|
> |w/o probability floor |Fail|
> |w/o reweighting mechanism |Fail|
>
> *Fail: not absorbed within 100 steps. KL loss=0.001 and KL penalty coefficients=0.01.*
>
> **Findings:**
> - OnePO still works under KL constraints.
> - Both the probability floor and reweighting are indispensable, which suggests that CLIP mechanism and gradient starvation are important obstacles to off-policy learning.
>
> To clarify, following DAPO, we **omit all KL constraints** in all our experiments (`Section 4.1`).
>
> > **W3. The unclear token-level claim.**
>
> We agree that the term “token-level” was imprecise and will revise it. The key distinction is:
>
> - **SFT**: Identically maximizes probabilities of every token.
>
> - **OnePO**: OnePO is **probability-aware**. It assigns stronger learning signals to low-probability tokens (below the floor $c$), while retaining the standard RL objective for the others.
>
> We will revise `token-level SFT` to `probability-aware token-wise optimization`.
>
> > **W4. Unclear specification of teacher trajectories.**
>
> We will clarify this. In our experiments:
>
> - teacher trajectories are **generated by external models**;
> - we use **one off-policy response per prompt** from a external model (e.g., GPT-5-Chat).
> - SFT, SFT+RL, and OnePO use the **same teacher source** for fairness.
>
> > **W5. Unclear definition of “Thinking”.**
>
> We agree that this terminology is ambiguous and will revise it.
>
> Our distinction is simply whether the teacher output contains an explicit reasoning segment before the final answer (e.g., `<think> ... </think> answer`). We will replace `thinking model` with `model with thinking trajectory`.
>
> > **W6. Rapid retirement of the weak teacher**
>
> We understand this concern. This behavior is expected and beneficial.
>
> **Why expected.**
>
> We use only **one teacher response per prompt**, while the on-policy side samples **8 rollouts**. When the teacher is weaker, the best on-policy rollout can surpass it early, so a high retirement ratio naturally emerges.
>
> **Why beneficial.**
>
> Teacher Retirement is designed to keep teacher guidance only while it remains useful. Surviving teacher trajectories (~25%) still provide useful guidance, performing close to the strong-teacher setting (`66.0 vs. 67.2` on HealthBench). Moreover, *Figure 5* shows that looser retirement thresholds (e.g., `Mean` or `No Retirement`) hinder learning.
>
> > **W7 & W8. Unclear details of benchmark split and rubric construction.**
>
> We apologize for these. For IFEval and GSM8K, we report **0-shot** results on the official test sets and do not use these benchmarks for training. Appendix D already includes a concrete rubric example and its design process. We will clarify this in the paper.
>
> > **W9. Stability of rubric-based rewards and missing convergence curves.**
>
> Agree. We will include convergence curves of the rubric-based reward, which are instance-specific but fixed during training, in the next version (partial data shown below):
>
> |Train step|0|20|40|60|80|100|120|
> |---|---:|---:|---:|---:|---:|---:|---:|
> |Open-end|0.34|0.69|0.69|0.70|0.70|0.75|0.77|
> |Close-end|0.29|0.38|0.48|0.52|0.55|0.58|0.59|
>
> **Observation.**
> Open-ended reward improves smoothly. We attribute this to two factors: (1) rubric-based reward is denser than simple exact-match reward, and (2) the rubrics, though instance-specific, provide a consistent optimization signal for improving medical response quality.
>
> We thank the reviewer again and will incorporate these clarifications in the revised paper.

---

> > ### Author Rebuttal · Reviewer_ff1Q · 2026-04-03
> >
> > I am still unclear on the KL divergence in PPO; "To clarify, following DAPO, we omit all KL constraints in all our experiments (Section 4.1)." - This is convincing.
> >
> > However, I am still unclear and not convinced (conceptually) with the experiment presented to back the prior claims on KL divergence.

---

> > > ### Author Response · Authors · 2026-04-06
> > >
> > > We sincerely thank the reviewer for the helpful follow-up and positive evaluation of our work. We are encouraged that our previous response addressed part of the concern. We appreciate the opportunity to further clarify the role of KL constraints, which we may not have explained with sufficient detail earlier.
> > >
> > > > **(1) How KL affects OnePO, conceptually**
> > >
> > > In PPO-style RL, KL constraints are typically introduced in two ways:
> > >
> > > 1. **KL regularization in the objective.** Add KL directly to the RL objective ($J_{\mathrm{RL}}$):
> > > $$
> > > J_{\mathrm{KL\mbox{-}loss}}(\theta)=J_{\mathrm{RL}}(\theta)\color{blue}{-\beta\cdot\mathrm{KL}(\pi_\theta\|\pi_{\mathrm{old}})}
> > > $$
> > >
> > > 2. **KL penalty in reward.** Add a KL penalty term to the trajectory reward ($R(\tau)$):
> > > $$
> > > R_{\mathrm{KL}}(\tau)=R(\tau)\color{blue}{-\beta\sum_{t=1}^{|\tau|}\log\frac{\pi_\theta(\tau_t \mid q,\tau_{<t})}{\pi_{\mathrm{old}}(\tau_t \mid q,\tau_{<t})}}
> > > $$
> > >
> > > In both cases, KL serves as an **additional regularization term** `[1]`: it limits how far the policy moves from a reference policy, **but is not intended to alter the primary learning target of RL**. Following recent GRPO-style studies, we omit KL in our main experiments because it can hinder exploration `[2]`.
> > >
> > > By contrast, **OnePO is designed to improve the RL objective,  rather than simply regularizing policy updates**. Specifically, it makes RL better absorb low-probability tokens from external off-policy trajectories. It addresses two bottlenecks:
> > >
> > > | Bottleneck|Why it happens|OnePO solution |
> > > |-|-|-|
> > > | **CLIP saturation**|For low-probability off-policy tokens, the ratio $r=\pi_\theta/\pi_{\theta_{\text{old}}}$ quickly exceeds the clipping range, so updates are clipped too early and gradients vanish.|The **Probability Floor $c$** prevents premature clipping. |
> > > | **Gradient starvation**|Beyond clipping, the RL gradient becomes negligible when $\pi_\theta^{(i,t)}\approx 0$ for low-probability tokens on off-policy trajectories (see **Appendix F.3**).|**Reweighting Term $g_{i,t}(\theta)$** turns the update into an MLE-like form, $A_i\nabla_\theta \log \pi_\theta^{(i,t)}$. |
> > >
> > > In short, **KL regularizes policy movement**, while **OnePO improves token learnability within RL**. The two are compatible, but they play different roles.
> > >
> > > > **(2) What the KL pilot experiment is intended to show**
> > >
> > > We apologize that our previous KL experiment may not have been explained clearly enough due to the rebuttal length limit.
> > >
> > > This experiment extends the first pilot study in **Appendix B.1** and asks:
> > >
> > > **Can the model absorb the target knowledge (`AHA-Medicine`) from off-policy trajectories during RL training, and how many training steps are needed?**
> > >
> > > For clarity, we present the results in two tables.
> > >
> > > - **Table 1. Off-policy token absorption**
> > >
> > > | Method|Step 1|Step 2|Step 3|Step 4|Step 5|Step 6|Step 7|Step 8 |
> > > |-|-:|-:|-:|-:|-:|-:|-:|-:|
> > > | OnePO|-6.974|-6.149|-4.223|-2.761|-1.026|-0.284|$\color{red}{-0.016}$|$\color{red}{-0.012}$ |
> > > | OnePO w/ KL loss + KL penalty|-6.974|-6.610|-5.510|-3.555|-2.258|-1.087|-0.373|$\color{red}{-0.041}$ |
> > > | OnePO w/o Reweighting|-6.974|-7.111|-7.116|-7.127|-6.935|-6.974|-6.540|-6.642 |
> > >
> > > *`Step` denotes the training step. The value is*
> > > $$
> > > -\frac{1}{t_e-t_s+1}\sum_{t=t_s}^{t_e}\log \pi_\theta(\tau_t \mid q,\tau_{<t}),
> > > $$
> > > *which is the average negative log-probability of the knowledge span `AHA-Medicine` on the off-policy trajectory. Values closer to 0 indicate better knowledge absorption. $\color{red}{\text{Red}}$ highlights indicate strong knowledge absorption.*
> > >
> > >
> > > - **Table 2. KL divergence during training**
> > >
> > > | Method|Step 1|Step 2|Step 3|Step 4|Step 5|Step 6|Step 7|Step 8|$\color{blue}{\text{avg. KL}}$ |
> > > |-|-:|-:|-:|-:|-:|-:|-:|-:|-:|
> > > | OnePO|0.000|0.065|0.061|0.028|0.004|0.005|0.007|0.006|0.023 |
> > > | OnePO w/ KL loss + KL penalty|0.000|0.041|0.024|0.007|0.013|0.002|0.003|0.002|$\color{blue}{0.012}$ |
> > > | OnePO w/o Reweighting|0.000|0.053|0.059|0.076|0.116|0.146|0.147|0.178|0.096 |
> > >
> > > *`Step` denotes the training step. The value is $\mathrm{KL}(\pi_\theta\|\pi_{\mathrm{old}})$, i.e., the KL divergence computed from rollouts. It measures how far the updated policy deviates from the reference policy. The $\color{blue}{\text{Blue}}$ highlight indicates that KL constraints reduce policy drift.*
> > >
> > > These tables are meant to show two points:
> > >
> > > 1. **KL acts as a regularizer:** It reduces KL divergence and keeps the policy closer to the reference policy.
> > > 2. **KL does not change OnePO’s core objective:** Even with KL constraints, OnePO still enables the model to absorb external off-policy knowledge, because its main contribution is to improve the RL objective itself.
> > >
> > >
> > > > **Reference**
> > >
> > > [1] On the Design of KL-Regularized Policy Gradient Algorithms for LLM Reasoning.
> > >
> > > [2] DAPO: An Open-Source LLM Reinforcement Learning System at Scale.
> > >
> > >
> > > **We sincerely thank the reviewer again for raising this important point.  We will revise the paper to add a more explicit discussion of KL regularization.**

---

### Official Review · Reviewer_1j7y · 2026-03-12

**Soundness:** 2
**Presentation:** 3
**Significance:** 2
**Originality:** 2
**Overall Recommendation:** 4
**Confidence:** 3

**Summary:**

This paper proposes OnePO, a SFT-free one-stage policy optimization method, to tackle the problems of distribution contraction and high training overhead in the traditional two-stage SFT+RL paradigm for LLM domain adaptation. It features two core mechanisms—Adaptive Objective Evolution and Teacher Retirement—that allow LLMs to selectively absorb off-policy knowledge directly in the RL phase while evading anchoring to off-policy distributions. Validated on the medical domain with just 20K samples using the Qwen3-8B-Base model, OnePO delivers competitive performance on medical benchmarks like HealthBench, demonstrating the feasibility of SFT-free one-stage RL for domain adaptation. The abstract clearly states the core research problem, methodological innovations, experimental validation and key conclusions, yet it omits any discussion of the method’s generalizability to other domains.

**Compliance With Llm Reviewing Policy:**

Affirmed.

**Final Justification:**

Conditional Acceptance Recommended

This paper proposes OnePO, an innovative method that breaks the traditional two-stage SFT+RL paradigm for LLM domain adaptation. Its core mechanisms are ingeniously designed and highly synergistic; the rigorous experimental validation on the medical domain yields competitive results, accurately addressing critical pain points such as distribution contraction and high training overhead in domain adaptation. The work features remarkable originality and practical application value, with clear presentation and good experimental reproducibility.

However, the paper has key unresolved limitations: it is only validated in the medical domain, with unproven cross-domain generalizability; all experiments are based on an 8B-scale model, lacking scalability analysis for large-parameter models; the theoretical derivation of core mechanisms is superficial, with no clear theoretical basis for hyperparameter selection. While the authors’ rebuttal fully addresses the issue of preserving the model’s general capabilities and provides rational analyses of cross-domain challenges and hyperparameter selection, it offers no supplementary experimental or theoretical evidence, failing to thoroughly resolve the aforementioned critical shortcomings.

Revisions required: 1) Add pilot validation experiments in at least one other professional domain (e.g., law/finance); 2) Conduct scalability tests on medium-to-large parameter models and provide relevant optimization notes; 3) Complete theoretical derivation for core hyperparameters and define their practical optimal ranges; 4) Supplement discussions on cross-domain generalizability in the abstract and introduction. The completion of these revisions will significantly enhance the paper’s academic value and practical deployment potential, meeting the requirements for formal acceptance.

**Key Questions For Authors:**

1. Has OnePO been validated in professional domains other than medicine (e.g., law, finance)? If not, what potential challenges do you anticipate for its cross-domain generalization, and are any mechanism adjustments needed?
2. Is there a theoretical optimal range or selection criterion for the probability floor *c* in Adaptive Objective Evolution, and does its optimal value vary across different tasks and domains?
3. How does OnePO inherently preserve the model’s general capabilities without degradation during training? Is this a result of specific mechanism design or the natural training characteristics of the one-stage paradigm?

**Limitations:**

OnePO is only validated in the medical domain, with no tests in other professional fields (e.g., law, finance), leading to unproven cross-domain generalizability.
All experiments rely on the 8B-scale Qwen3-8B-Base model, lacking scalability analysis on larger-parameter models (e.g., 70B, 130B).

**Strengths And Weaknesses:**

Strengths
1. The research targets critical and pressing pain points in LLM domain adaptation. It accurately identifies two fundamental flaws of the classic SFT+RL two-stage paradigm: blind cloning that restricts subsequent exploration, and increased training complexity plus stringent data quality demands from multi-stage pipelines. It also pinpoints the limitations of pure and mixed-policy RL in off-policy knowledge learning without SFT cold start, making the research highly valuable and targeted.
2. The OnePO paradigm boasts notable methodological innovation. Breaking away from the traditional two-stage training framework, its two core mechanisms are cleverly designed and synergistic. Adaptive Objective Evolution enables a smooth shift from guided imitation to autonomous exploration, while Teacher Retirement prevents off-policy anchoring and reduces sensitivity to off-policy data quality, well balancing off-policy knowledge absorption and autonomous exploration.
3. The experimental validation is rigorous and convincing. The authors constructed a dedicated 20K medical dataset (OneData-Health-20K), designed targeted pilot studies, and set up comprehensive baselines and ablation experiments. These fully verify the effectiveness of OnePO and its core components, and the excellent experimental results on medical benchmarks further prove the method’s practicality and efficiency.

Weaknesses
1. Cross-domain generalizability is insufficiently validated. The method is only tested in the medical domain, with no validation on other professional fields (e.g., law, finance) or general tasks. This leaves the method’s applicability across different domains unproven, and the impact of domain-specific characteristics on its performance remains unclear.
2. There is a lack of scalability analysis for larger model sizes. All experiments are conducted solely on the 8B-scale Qwen3-8B-Base model, with no tests on larger-parameter models (e.g., 70B, 130B). The training stability, efficiency and performance of OnePO on large models are unknown, which limits the method’s practical deployment potential in real-world scenarios.
3. The theoretical analysis of core mechanisms is shallow. Although experimental results confirm the effectiveness of Adaptive Objective Evolution and Teacher Retirement, there is a lack of in-depth theoretical derivation—such as the optimal range of the probability floor *c* and the theoretical basis for the Teacher Retirement threshold selection. This reduces the theoretical depth and interpretability of the proposed method.

---

> ### Author Rebuttal · Authors · 2026-03-24
>
> Dear Reviewer 1j7y,
>
> Thank you very much for your time and effort in reviewing our work.
>
> We noticed that the current review text mentions "**PreCoMem**", which seems to differ from our submission "**OnePO**". We respectfully suspect that a technical error may have occurred during the pasting process.
>
> Could you please help us check if the review was intended for another manuscript? We are very eager to receive your professional feedback on OnePO so that we can address any concerns and improve our work accordingly.
>
> Thank you for your assistance and for your valuable time.
>
> Best regards,
>
> The Authors

---

### Decision · Program_Chairs · 2026-04-30

**Decision:**

Accept (regular)

**Comment:**

This paper proposes an SFT-free, one-stage policy optimization method, OnePO. Reviewers agreed that the paper is clearly presented, addresses an important problem in LLM domain adaptation, and introduces a well-designed method with good empirical performance.

The rebuttal addressed several concerns by providing additional results on new domains, hyperparameter analyses, robustness evaluations across graders, and a human validation study.

However, a few key limitations remain. In particular, reviewers noted the lack of scalability analysis for larger models and the relatively superficial theoretical derivation of the core mechanisms. In addition, the experimental evidence supporting the claims on KL divergence was found to be insufficiently convincing.

For the final version, the following revisions are required:
(1) Include pilot validation experiments in additional professional domains (e.g., law or finance);
(2) Conduct scalability tests on medium to large-parameter models and provide corresponding optimization notes;
(3) Strengthen the theoretical derivation of key hyperparameters and clarify their practical optimal ranges;
(4) Expand the discussion of cross-domain generalizability in the abstract and introduction.